# Epigenetically driven and early immune evasion in colorectal cancer evolution

Eszter Lakatos [1,2,9] ✉, Vinaya Gunasri[3,4,9], Luis Zapata[1], Jacob Househam [1], Timon Heide [1,5], Nicholas Trahearn [1,3], Ottilie Swinyard[4], Luis Cisneros [6], Claire Lynn[1], Maximilian Mossner[1,4], Chris Kimberley[4], Inmaculada Spiteri [1], George D. Cresswell [1,7], Gerard Llibre-Palomar [4], Miriam Mitchison[8], Carlo C. Maley [6], Marnix Jansen [3], Manuel Rodriguez-Justo [3], John Bridgewater[3], Ann-Marie Baker [1,4], Andrea Sottoriva [1,5,10] ✉ & Trevor A. Graham [1,4,10] ✉

Immune system control is a principal hurdle in cancer evolution. The temporal dynamics of immune evasion remain incompletely characterized, and how immune-mediated selection interrelates with epigenome alteration is unclear. Here we infer the genome- and epigenome-driven evolutionary dynamics of tumor-immune coevolution within primary colorectal cancers (CRCs). We utilize a multiregion multiomic dataset of matched genome, transcriptome and chromatin accessibility profiling from 495 single glands (from 29 CRCs) supplemented with high-resolution spatially resolved neoantigen sequencing data and multiplexed imaging of the tumor microenvironment from 82 microbiopsies within 11 CRCs. Somatic chromatin accessibility alterations contribute to accessibility loss of antigen-presenting genes and silencing of neoantigens. Immune escape and exclusion occur at the outset of CRC formation, and later intratumoral differences in immuno-editing are negligible or exclusive to sites of invasion. Collectively, immune evasion in CRC follows a 'Big Bang' evolutionary pattern, whereby it is acquired close to transformation and defines subsequent cancer-immune evolution.

Tumors are shaped continuously by interactions with their environment, especially by the ongoing 'war' with the immune system. Neoantigens—new peptides originating from somatic mutations—may elicit immune recognition, but tumors ultimately evade immune elimination by immune-editing (losing antigenic mutations), immune-escape (for example, hampering antigen presentation), immune exclusion (manipulating the tumor microenvironment (TME) to limit immune presence) or a combination of these mechanisms[1]. Immune evasion is unlikely to be a binary on/off trait, but rather a continuous phenotype modulated by the strength of the contribution of these factors. Immunotherapies aim to re-engage the immune system in its war against cancer; thus, understanding the mechanisms underlying immune evasion is paramount for treatment success.

Colorectal cancers (CRCs) generally display a relatively active immune microenvironment with substantial tumor-infiltrating lymphocytes[2]. About 15% of CRCs are mismatch repair (MMR) deficient (MMRd), which is associated with a higher neoantigen burden, enhanced immune presence[3,4] and good response to immune checkpoint blockade (ICB) therapies; however, up to 30% of MMRd CRCs do not respond[5]. MM proficient (MMRp) CRCs harbor a lower mutation burden than MMRd CRCs and ICB treatments are ineffective[6], indicating that immune evasion in MMRp tumors probably occurs through mechanisms alternative to those targeted by current ICB drugs. Nonetheless, immune infiltrate is prognostic for CRC[7], indicating a central role for the immune system in CRC evolution. Indeed, most MMRp cancers carry (several) putative clonal neoantigen mutations[8] and,

---

whereas only a subset of these might be sufficiently presented[9], they may still be capable of T cell activation[10–12].

Immune evasion through genetic means (for example, mutations that hamper immune recognition of neoantigens) is common in CRCs[4,13]. Our previous mathematical modeling found that escape is essential for MMRd CRC development, and is a crucial step in MMRp CRC formation when immune surveillance is stringent[4,8,14]. The role of the epigenome in modulating tumor antigenicity has received less attention: somatic changes in chromatin organization are known to contribute to the cancer immunophenotype, but their impact on immune escape and editing have not been completely assessed[15–17]. In seminal studies from the TRACERx consortium[18,19], promoter hypermethylation has been identified as mechanism of neoantigen silencing in lung cancer[20]. Analogously, closing chromatin may lead to repression of antigen expression, meaning that immune selection could strongly shape a cancer's epigenome architecture.

Previous analyses relied on bulk sequencing data from superficial tumors, and so were probably unable to detect subclonal and heterogeneous immune evolutionary dynamics that could be associated with disease spread. Further, the invasive margin—the cancer region in direct contact with normal tissue—is probably a main determinant of overall immune evasion/elimination, but has rarely been studied directly because it is typically fixed for diagnostic purposes and unavailable for research[21,22]. In summary, spatial heterogeneity in CRC immuno-editing and escape remains incompletely characterized.

Here we explore the interplay between immune evasion and the epigenome, highlighting a role for chromatin architecture in suppressing expression of neoantigens and antigen-presenting machinery in CRC. We characterize immune evasion (microenvironment restructuring and genetic immune escape) at the individual tumor gland-level, exploring intratumor heterogeneity within and across different morphological contexts. We leverage our existing multiregion multiomic sequencing (matched genome, transcriptome and chromatin accessibility profiling) of 495 single glands (representing 29 CRCs) from our previously published Evolutionary Predictions in Colorectal Cancer (EPICC) study[15,23]. We supplemented this with newly generated data that combines targeted neoantigen sequencing and highly multiplexed TME profiling of 82 microbiopsies representing distinct tumor-associated regions from a subset of 11 EPICC cases, including lymph node metastases and distant normal mucosa.

## Results

### Multimodal immune analysis of CRC microbiopsies

We analyzed the immune landscape of 29 CRCs at single tumor gland resolution, using multiomic (whole-genome sequencing (WGS), RNA sequencing (RNA-seq) and assay for transposase-accessible chromatin using sequencing (ATAC-seq)) analysis of fresh frozen (FF) single glands, coupled with high-depth panel sequencing (PS) and cyclic immunofluorescence[24] (CyCIF) imaging of matched formalin-fixed paraffin-embedded (FFPE) biopsies. The collection and processing of FF samples have been described previously[15,23] (Fig. 1a). We obtained diagnostic FFPE blocks from 11 patients with MMRp stage III CRCs and lymph node metastasis. We microdissected small regions representing superficial tumor, invasive margin and lymph node deposits (Fig. 1a) and sequenced genomic regions associated with the immunopeptidome[25] (Methods). In a subset of these regions, we visualized 22 proteins that identified immune cell types or regulatory receptor expression (Supplementary Table 2) using CyCIF (Methods). In total, we assembled sequencing data from a total of 495 FF and 82 FFPE biopsies from 29 patients (median of 15 FF and eight FFPE biopsies per patient) with concurrent CyCIF analysis of eight patients.

In both whole-genome- and panel-sequenced biopsies, we predicted the antigenicity of each somatic mutation using NeoPred-Pipe[26]. We defined the (proportional) neoantigen burden of each biopsy as the number of mutations present that gave rise to at least one strong-binding, cancer-specific mutated peptide, normalized to the total burden of protein-changing mutations. For samples with sufficient mutations, we quantified immune selection using the ratio of nonsynonymous to synonymous mutations in the immunopeptidome (immune dNdS) using SOPRANO[25] (Fig. 1a; Methods).

In FF−WGS samples, we characterized candidate immune escape alterations: single nucleotide variants (SNVs), frameshifts and loss-of-heterozygosity (LOH) in *HLA* genes, high-impact alterations in other antigen-presenting genes (APGs; see Methods for list) and high expression of PD-L1 or CTLA-4 (as measured by RNA-seq). Samples with a LOH, frameshift, stop-gain alteration or several SNVs in APGs, or with overexpression of PD-L1 and CTLA-4 were labeled 'escaped,' whereas samples with a single SNV or moderate PD-L1/CTLA-4 expression were labeled 'potential escape' (Fig. 1b and Extended Data Fig. 1).

Neoantigen burden, immune dNdS, genetic immune escape alterations and immune markers each showed high interpatient but moderate intrapatient variability (Fig. 1b,c). As expected, total mutation burden was correlated strongly with the number, but not the proportion of antigenic mutations; immune dNdS values showed a reversed, albeit highly uncertain relationship with proportional burden (Extended Data Fig. 2). MMRd cancers (n = 6) had significantly higher mutation load than MMRp cancers (Extended Data Fig. 2a,b; median synonymous burden 307 versus 18; median total neoantigen burden 292 versus 23, two-sided Wilcoxon signed-rank test $P < 10^{-16}$) and all carried clonal high-impact immune escape alterations (Fig. 1b and Extended Data Fig. 1). Overall, immune escape alterations were detected in 14 of 29 cancers; these were clonal in 8 cases and subclonal in 6 (all MMRp; Extended Data Fig. 1).

### Epigenetic regulation of antigen presentation

We showed previously that somatic chromatin accessibility alterations (SCAAs) evoked genome-wide rewiring of transcription factor (TF) binding that altered interferon signaling, suggesting suppression of immune signaling through epigenetic regulation[15]. We therefore examined the role of the epigenome in enabling immune escape by looking for SCAAs associated with APGs (Methods) in 25 CRCs. We detected a total of 45 SCAAs, all in promoter regions upstream of these APGs. Of these 34 APGs, 21 (62%) had at least one SCAA and 9 of 25 (36%) of patients had at least one APG affected by a SCAA. Notably, 42 (93%) of the SCAAs were losses of accessibility, significantly different than expected based on the genome-wide distribution of SCAAs (P = 0.025; Methods). None of the SCAAs co-occurred with somatic mutations in those genes (Fig. 2a). This exclusivity could indicate that mutations are more likely to be found in expressed genes or that SCAAs are an alternate route to antigen presentation disruption.

We explored predicted TF binding sites within SCAA-loss APG promoter regions. A total of ten ATAC-seq peaks in these promoters showed recurrent somatic loss (present in more than one patient), associated with eight distinct APGs. We found ten TFs that bound more than two of these regions (Methods), most notably NFIC, which had binding sites in the silenced promoter of all eight recurrently SCAA-affected APGs (Extended Data Fig. 3a). Although these TFs had a high number of binding sites genome-wide, they were enriched for APG-SCAA-loss site binding (one-sided Fisher-test P = 0.042). Furthermore, they covered several APG promoters, whereas other TFs targeted few APGs (for example, ZNF32 bound only the *ERAP2* SCAA-loss sites; Supplementary Table 1). Except for CTCFL and TBPL2, these TFs were expressed in most RNA sequenced samples, confirming their relevance in CRC.

Examining gene expression of SCAA-loss APGs, we observed that ERAP2 (SCAA-loss in three cancers and three adenomas), had considerably lower expression in the affected cancers than in normal tissue (mean variance stabilizing transformation (VST) count = 6.8 versus 7.9) and in unaffected cancers (mean VST count = 8.1; Extended Data Fig. 3b,c). However, we did not observe a systematic decrease in expression across all gene–patient combinations (Extended Data Fig. 3c). Looking more

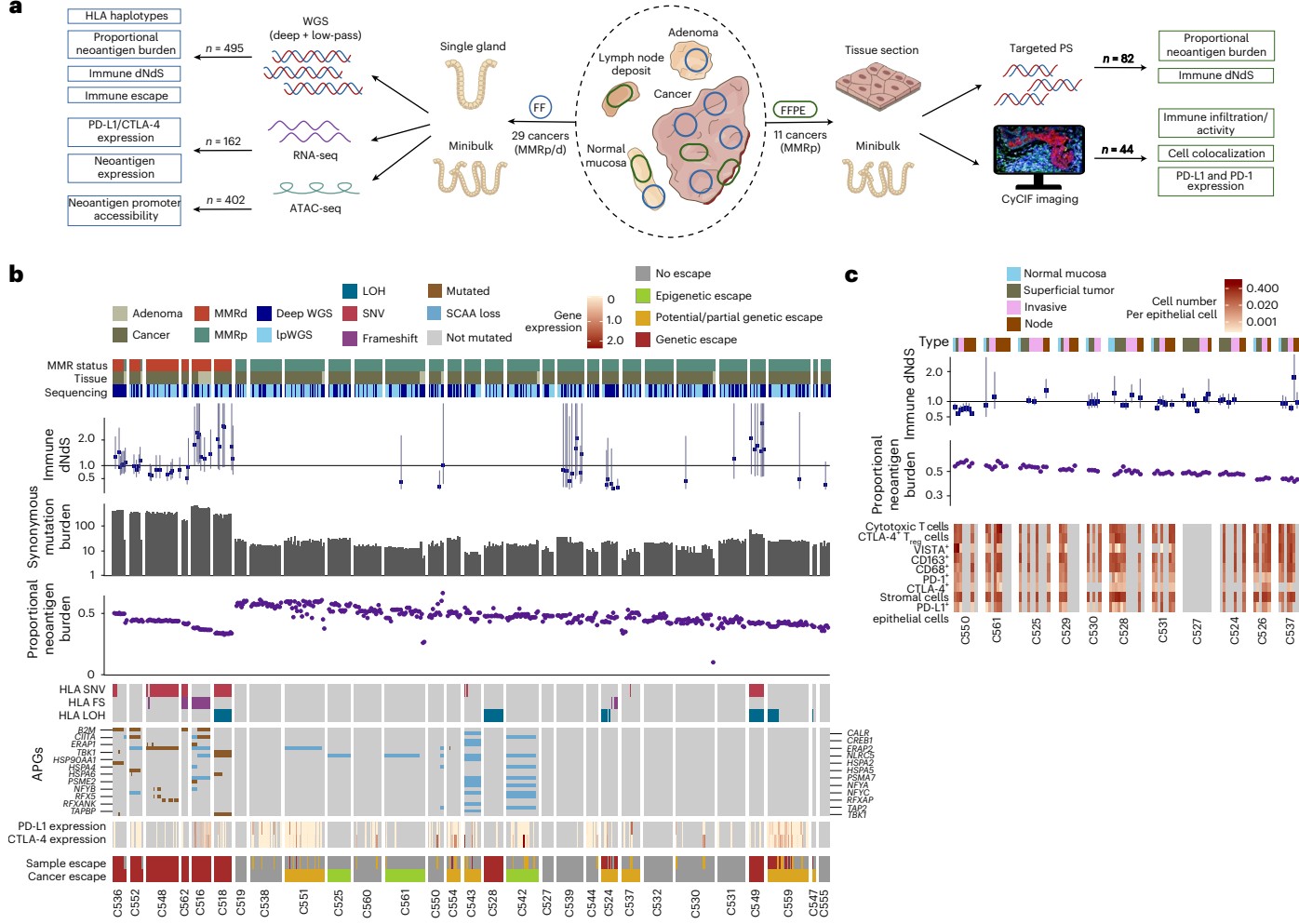

**Fig. 1 | Multiomic sequencing and multiplex imaging of the cancer-immune landscape of CRCs. a**, Overview of sample collection (middle), processing and analysis of FF (left) and FFPE (right) biopsies. **b**, Neoantigen burden, synonymous mutation burden, immune dNdS and immune escape mutation overview of all FF–WGS samples, ordered by average neoantigen burden and MMR status. Error bars depict CIs for immune dNdS estimates, truncated at 3.5. **c**, Neoantigen burden, immune dNdS and immune infiltrate overview of all FFPE–PS samples, ordered by average neoantigen burden.

broadly, chromatin accessibility of promoters (quantified as ATAC-seq counts per million reads, regardless of somatic status) showed a positive correlation with gene expression across all genes (Spearman correlation, $R = 0.12$; $P < 10^{-16}$), which was more pronounced in genes with low average expression (gene group 4 in ref. 23; $R = 0.19$). We suggest SCAAs generally impact gene expression but, due to their permissive (not directly causative) nature, the effect(s) may not always manifest. However, chromatin changes may contribute to phenotypic plasticity by extending or restricting the range of accessible gene expression states.

Motivated by our previous findings that gene expression in CRCs shows high plasticity[23], we examined the heritability of APG expression (Supplementary Note) using phylogenetic signal analysis[27,28]. Strong or recurrent phylogenetic signal correlated with subclonal HLA LOH and subclonal differences in the pattern of open chromatin in cancer C559 (Extended Data Fig. 3d,e). However, the generally low level of phylogenetic signal (only in 17 of 297 gene–cancer combinations evaluated), combined with high intratumor heterogeneity confirms that the expression of APGs is plastic, similar to most genes.

## Immuno-editing through SCAAs and transcriptional regulation

We examined whether epigenome reorganization preferentially silences neoantigens, (that is, epigenetic immuno-editing). We collated the chromatin accessibility of all genetic loci where a protein-changing

mutation was detected. Neoantigens were enriched in genes where SCAAs tended to close chromatin (Fisher's exact test odds ratio $(OR)_{(neoantigen\ and\ SCAA\ loss)} = 1.46\ (1.05–2.03)$; Fig. 2b), and this observation was also true on the individual cancer level ($P = 0.017$; Fig. 2c). SCAA loss enrichment was significant in MMRd cancers (Fisher's exact test $OR_{(neoantigen\ and\ SCAA\ loss)} = 1.52\ (1.03–2.27)$) but not in MMRp—the latter probably due to lack of power, as the distribution of neoantigens and SCAA-loss-affected mutations was not different between MMRp and MMRd cancers ($\chi^2$-test $P = 0.37$). We also found that significantly more SCAA losses were associated with neoantigens than with nonantigenic SNVs ($P = 0.0085$; Extended Data Fig. 4a).

Next, we examined transcriptional expression of neoantigens. We compared the proportion of neoantigens to nonantigenic protein-changing mutations falling within specific genes, and found that genes with high and consistent expression (groups 1 and 2 of ref. 23; Supplementary Note) were depleted significantly of clonal, but not subclonal, SNV neoantigens (Fisher's exact test $OR_{(neoantigen\ and\ in\ group\ 1)} = 0.73\ (0.54–1.00)$ and $OR_{(neoantigen\ and\ in\ group\ 2)} = 0.75\ (0.60–0.94)$; Fig. 2d and Extended Data Fig. 4b–d). Frameshift neoantigens showed similar patterns (Extended Data Fig. 4e–h), suggesting clonal neoantigens are typically found in genes with low and/or variable expression.

We also examined allele-specific expression of neoantigens. Clonal neoantigens were significantly less likely to be expressed (Fisher's exact test $OR_{(neoantigen\ and\ not\ expressed)} = 1.53\ (1.03–2.30)$) than non-antigenic

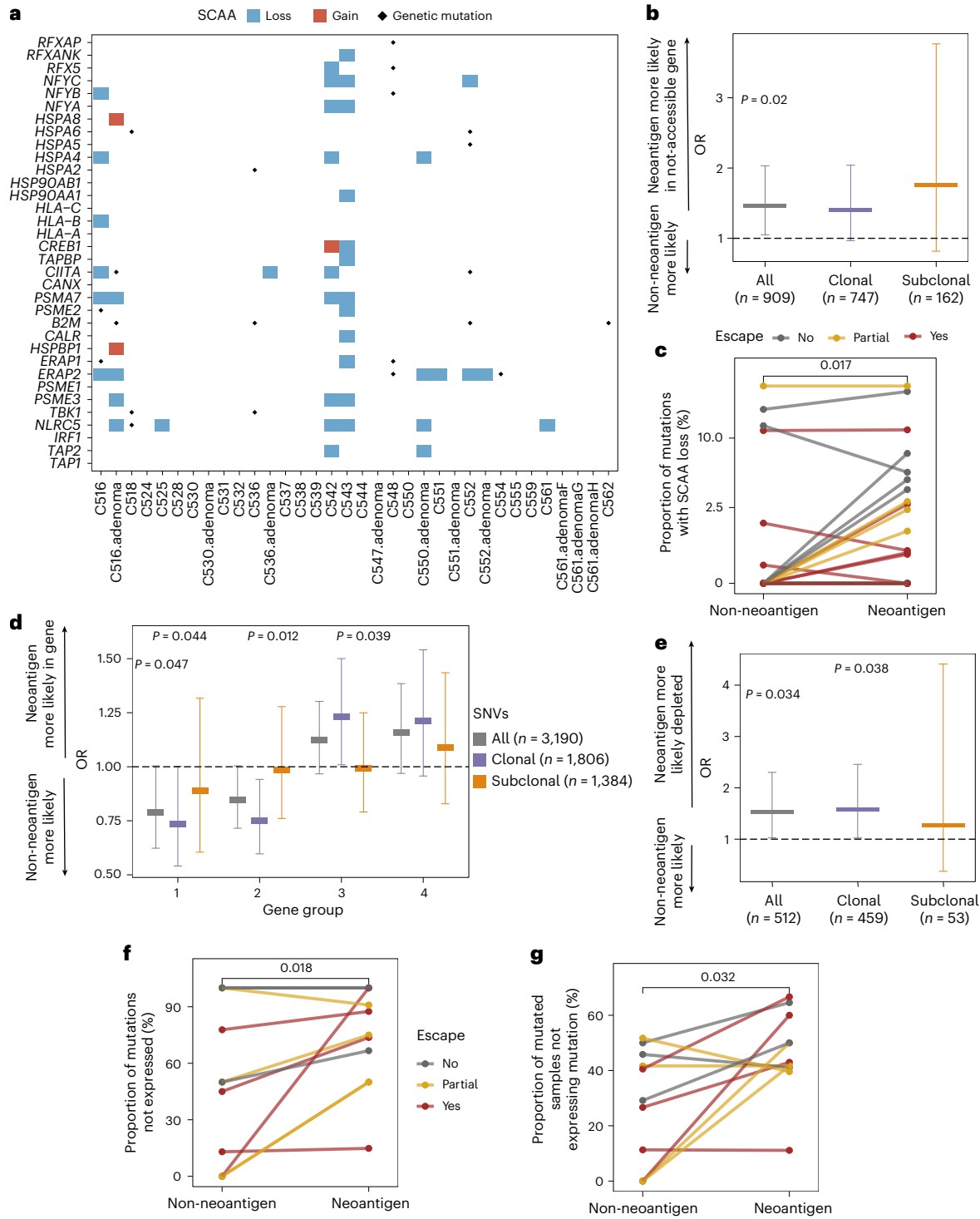

**Fig. 2 | SCAAs and transcriptional regulation decrease neoantigen presentation. a**, Heatmap of SCAAs in APGs. Blue and red rectangles show SCAA losses and gains in gene-associated promoter regions, with dots indicating somatic mutations. **b**, Two-sided OR of a neoantigen versus nonantigenic mutation being located in a gene affected by promoter SCAA loss. **c**, Proportion of mutations identified by WGS that are located downstream of a SCAA loss promoter in *n* = 24 cancers. **d**, Two-sided OR of a neoantigen versus nonantigenic mutation being located in genes of gene groups 1–4 from ref. 23. All, clonal and subclonal mutations are shown separately. **e**, Two-sided OR of a neoantigen versus nonantigenic mutation being transcriptionally edited (present in

WGS but not expressed in RNA-seq). **f**, Proportion of mutations found in WGS that are edited in RNA for each FF–WGS cancer with matched RNA-seq data (*n* = 12). **g**, Proportion of samples within a cancer that carry a given (antigenic or nonantigenic mutation) in WGS but not in corresponding RNA-seq (*n* = 12 cancers). In **b**, **d** and **e**, markers indicate OR estimates with error bars showing the corresponding 95% CI. Significant *P* values from unadjusted two-sided Fisher's exact tests are indicated above each bar. In **c**, **f** and **g**, cancers are colored according to their immune escape status (same color key applies), and *P* values from two-sided paired Wilcoxon-tests are indicated on top of each panel.

clonal SNVs (Fig. 2e), and this was also true on the level of individual cancers (Fig. 2f). Subclonal mutations showed no significant difference; however, this could have been due to lack of power, as downsampling clonal mutations to similar numbers also eliminated the previously observed difference (only 3 of 50 downsampled clonal datasets significant). Further, neoantigen silencing was more widespread within a cancer (observed in more biopsies from that cancer) than silencing of nonantigenic SNVs (Fig. 2g). Overall, around 50% of neoantigens were found to be transcriptionally immuno-edited, with the exception of C516—a highly escaped Lynch syndrome-associated MMRd cancer (Extended Data Fig. 1). After exclusion of this cancer, we confirmed transcriptomic silencing of neoantigens in all (Fisher's exact test $OR_{(neoantigen \, and \, not \, expressed)} = 2.07$ (0.99–4.40), $OR_{(clonal \, neoantigen \, and \, not \, expressed)} = 2.88$ (1.2–7.0)) and MMRp cancers ($OR_{(clonal \, neoantigen \, and \, not \, expressed)} = 3.7$ (1.1–14.3)).

Collectively, we observed that (clonal) neoantigens are enriched in genes with low expression and/or in regions of closed chromatin and are further depleted through allele-specific expression modulation. All these processes of postgenetic immuno-editing are likely to have an underlying somatic epigenetic mechanism.

## Immune exclusion and suppression across tumor regions

We next examined how the structure and composition of the TME contributed to immune evasion. Quantitative analysis of CyCIF data showed that overall tumors were significantly depleted of immune cells (Fig. 1c). The fraction of CD8+ cytotoxic T lymphocytes (CTLs) was significantly lower in superficial tumor and in the invasive margin than in adjacent normal mucosa ($P = 3 \times 10^{-5}$ and $P = 3 \times 10^{-3}$, respectively; Fig. 3a) and CTLs were more distant from epithelial cells in tumor-containing regions than in normal mucosa ($P < 10^{-16}$; Fig. 3b,c). CTL–tumor cell distance was highest in superficial tumor regions. Further, CTLA-4-expressing FOXP3+ regulatory T cells (T_reg cells), associated with an immune-suppressive function, were enriched in all tumor regions compared to normal mucosa ($P = 0.012$, $P = 0.01$ and $P = 6 \times 10^{-3}$ for superficial tumor, invasive margin and node; Fig. 3d), consistent with previous reports[29].

We confirmed that lymphocyte infiltration was reduced in superficial tumor samples compared to adjacent and distant normal mucosa, using deep-learning-based cell classifier analysis of hematoxylin and eosin (H&E) staining[30] ($P = 1.5 \times 10^{-6}$, $P = 1.6 \times 10^{-3}$, respectively; Extended Data Fig. 5; Methods). However, the total lymphocyte count in the invasive margin and lymph node deposits was similar to that observed in normal samples, indicating specific depletion of CTLs at invasion.

## Prevalence and consequence of genetic immune escape

We next examined the evolution of genetic immune escape alterations. Clonal immune escape mutations were detected in 8 of 29 CRCs (Figs. 1b and 4a,b and Extended Data Fig. 1) with three cancers (C518, C524, C548; Fig. 4c,d and Extended Data Fig. 1) carrying multiple HLA alterations on independent branches of their phylogenetic tree. This parallel emergence suggests strong selection for immune escape[31]. Minor subclones with immune escape alterations (alteration present in less than 25% of samples from a tumor) were detected in a further four cases (C537, C543, C547, C559; Fig. 1b and Extended Data Fig. 1). Two HLA mutations (in C524 and in C537) were also confirmed in the invasive margin of matched FFPE–PS samples.

We found that alterations with the highest predicted impact on immune evasion (based on previous studies) were mostly shared across the whole tumor or several spatially distinct regions, meaning that they were early evolutionary events. For example, beta-2-microglobulin (*B2M*) mutations were all clonal ($n = 4$; Fig. 4b and Extended Data Fig. 1), and matched RNA-seq revealed significantly decreased *B2M* expression in these cancers, confirming the impact of these mutations (Extended Data Fig. 6a). Similarly, mutations in *NLRC5* and *RFXAP*—essential factors in the MHC class I enhanceosome[32]

that reduced expression of class I MHC genes (*HLA-A, -B* and *-C*; Extended Data Fig. 6b)—were also shared across several regions. HLA LOH, a common immune escape mechanism shown to be more impactful than *HLA* mutations[33,34], was also clonal or near-clonal in four out of five cancers with HLA LOH (Fig. 4a and Extended Data Fig. 1). In contrast, probably low-impact (SNV) mutations in APGs showed the highest variability within tumors: we detected eight *HLA* SNVs and five SNVs in other APGs that were unique to a single gland or tumor region. We suggest that these alterations probably confer only limited disruption to neoantigen presentation and are weakly selected.

To examine the timing of immune evasion during initial cancer formation, we evaluated neoantigens and immune escape variants in 25 glands from eight colorectal adenomas (CRAs), including an advanced cancer-adjacent MMRd adenoma from patient C516. We note that these adenomas were not the precursor to the CRC, evidenced by the very few shared CRA-CRC mutations. No adenoma glands carried immune escape mutations, with the exception of the advanced CRA in C516 (Extended Data Fig. 1). Further, CRAs had significantly lower proportional neoantigen burden than CRCs (Fig. 4e; Methods), indicating that immune surveillance is more active or effective in CRAs than in CRCs. The two MMRp CRAs detected adjacent to MMRd CRCs did not bias this observation, as it was reiterated in MMRp-only samples (Extended Data Fig. 6c). Therefore, we suggest that immune escape occurs at the outset of, or early during, CRC outgrowth.

We measured how the presence and extent of genetically or epigenetically driven immune escape correlated with immune selection and infiltration. In MMRp CRCs, we observed a trend for lower overall neoantigen burden (Fig. 4f) in genetically escaped regions, especially when compared to cancers with only epigenetic alteration, although this trend was not significant when accounting for same-patient biopsies through a mixed-effect model (Methods). Noteworthy was that C524 and C548—cases with parallel evolution of HLA alterations—had biopsies with immune dNdS <1 (stringent immuno-editing) suggesting past stringent immune surveillance could have led to selection for these HLA alterations (Fig. 1b). Partially escaped CRCs also had an intermediate overall proportional neoantigen burden, with immune dNdS values below 1, indicating immuno-editing (Extended Data Fig. 6d).

In the matched CyCIF data, PD-1+ cells, VISTA+ cells and CD163+ cells, which are known to mediate T cell exhaustion[35] and promote tumor growth, were enriched in partially escaped cancers (Fig. 4g and Extended Data Fig. 7a,b), indicating a TME with impaired immune elimination ability. Fibroblasts were more abundant in escaped cancers, while CTLs and T_reg cells showed no difference by escape status (Fig. 4h and Extended Data Fig. 7b,c). Epigenetically escaped cancers (without genetic escape), when compared to nonescaped biopsies, showed an enrichment of CD68+ cells (macrophages) and CD45RO+ cells (memory T cells), suggesting an association between SCAA-loss of antigen presentation and the TME. Using RNA-seq data, we explored the transforming growth factor beta (TGFβ) signaling pathway, often modulated in cancer-associated fibroblasts and associated with T cell exclusion. We observed a trend for higher fibroblast TGFβ signature (F-TBRS[36]) in partially and epigenetically escaped samples, and that TGFβ receptor 2 (TGFBR2) had significantly higher expression in escaped cancer biopsies than nonescaped or partially escaped ones (Extended Data Fig. 7g,h). These findings suggest that genetically escaped cancers were subject to strong selection during early development, before expansion or gaining subclonal escape.

We also assessed the intratumor distribution of neoantigens (Supplementary Note) and found that significant depletion in variant allele frequencies (VAFs)—associated with immune selection—was restricted to escaped MMRp cases (Kolmogorov–Smirnov test $P = 0.012$; Extended data Fig. 8a–c), and neoantigens were as likely to be shared between samples from a tumor as nonantigenic mutations (Extended Data Fig. 8d–f). These data are indicative of relatively uniform immune selective pressures across large tumor regions.

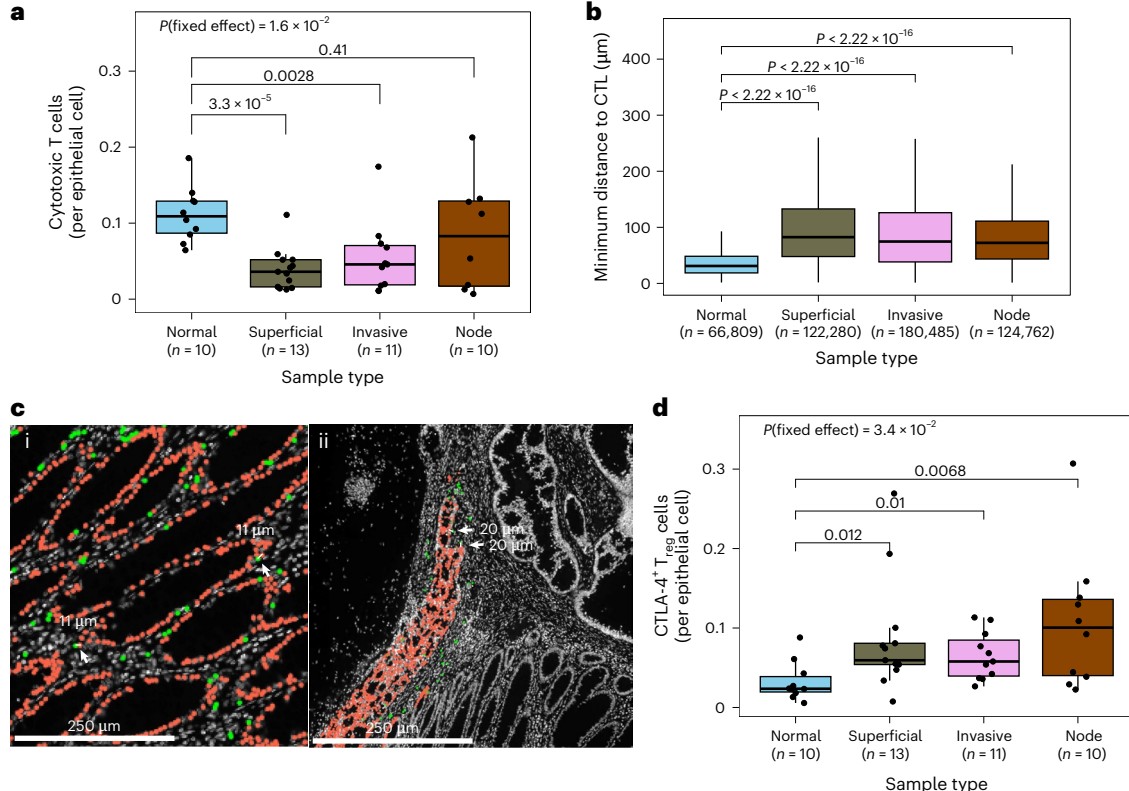

**Fig. 3 | Immune exclusion and clonal immune escape in CRCs. a,d**, Number of cytotoxic T cells (**a**) and CTLA-4$^+$ T$_{reg}$ cells (**d**) normalized to epithelial cells in each CyCIF image, grouped by sample type. **b**, Distance between epithelial cells and the closest cytotoxic T cell in different sample regions ($n$ = 66,809; 122,280; 180,485 and 124,762 in normal, superficial, invasive and node, respectively). **c**, Representative CyCIF images showing epithelial cells (red) and cytotoxic T cells (green) in normal mucosa (i) and in superficial tumor (ii). Points in **a** and **d** correspond to individual ROIs. The $P$ value of the mixed-effects model incorporating Patient as a random effect is indicated at the top of **a** and **d**.

## Intratumor heterogeneity in immuno-editing

We sought to further explore intratumor heterogeneity in genetic immuno-editing using our spatially resolved gland-level samples. We examined the correlation between histological region and neoantigen burden in comparison to the presence/absence of immune escape and other gland-to-gland tumor-intrinsic difference. The genetic similarity of samples correlated with the physical/phylogenetic distance between samples (Extended Data Fig. 9a,b). Multivariable regression (Methods) showed that the vast majority of neoantigen burden variation was explained by patient-specific effects and progression status (CRA versus CRC), with negligible contribution from other variables (Fig. 5a and Extended Data Fig. 9c). Patient-specific effects also dominated over the histological location of the sample (superficial, invasive edge or node) (Fig. 5b and Extended Data Fig. 9d). Using the immune dNdS measure in place of neoantigen burden gave analogous results (Extended Data Fig. 9e,f). This was also true when clonal and subclonal mutations were considered separately, and when tumors were stratified by immune escape status (Extended Data Fig. 9g–i).

We analyzed the impact of subclonal immune escape by comparing neoantigen burdens between mutated and unmutated phylogenetically close regions in cancers with subclonal immune escape mutations (Extended Data Fig. 1; Methods). Subclonal escape was not associated systematically with higher normalized neoantigen burden or immune dNdS values, with the exception of HLA-mutated biopsies in C543 that did carry a higher antigenic burden (Fig. 5c,d). These data confirm that subclonal escape had, at most, a modest effect on the intensity of immuno-editing.

To confirm that results were not dependent on the clonal neoantigen burden, we performed downsampling of clonal SNVs and repeated the multivariate regression against proportional neoantigen burden

(Supplementary Fig. 1 and Supplementary Methods). In most cases we still did not observe significant differences between subclonally immune escaped and phylogenetically related regions.

## Localized tumor-immune interactions at the invasive margin

We next focused on immuno-editing and TME structure in FFPE–PS samples, to identify spatially diverse determinants at the invasive margin and lymph node deposits. We mapped association between TME components and immuno-editing through a multivariable analysis extending Fig. 5b, where we included key infiltrates identified in our previous analyses (Extended Data Fig. 10a). We found that CTLs and CD163$^+$ macrophages and CTLA-4$^+$ T$_{reg}$ cells showed an association with neoantigen burden. When categorizing lymphocytes as tumor-infiltrating, adjacent or distant[37], we observed that invasive margins had a higher ratio of infiltrating lymphocytes compared to the superficial tumor (Fig. 6a), although this observation could not be reproduced in CyCIF images, probably due to reduced sample size. CyCIF revealed that the proportion of PD-L1$^+$ tumor cells (probably repressive of T cell activity) was significantly higher at the invasive margin than in superficial tumor (Fig. 6b). These results imply heightened immune surveillance at the invasive margin that is then reduced in node deposits.

We sought additional evidence of heightened immune surveillance at the invasive edge by deriving 15 cellular neighborhoods[38] (CNs), from our CyCIF data (Supplementary Methods and Extended Data Fig. 10b). The fraction of cells belonging to CN3 (highly enriched for PD-L1$^+$ cells, including PD-L1$^+$ epithelial cells) was significantly higher in invasive margin and node deposits than in superficial tumor, while CN10 (also with general enrichment for PD-L1$^+$ cells) was present at similar proportions in all tumor-associated regions (Extended Data Fig. 10c,d).

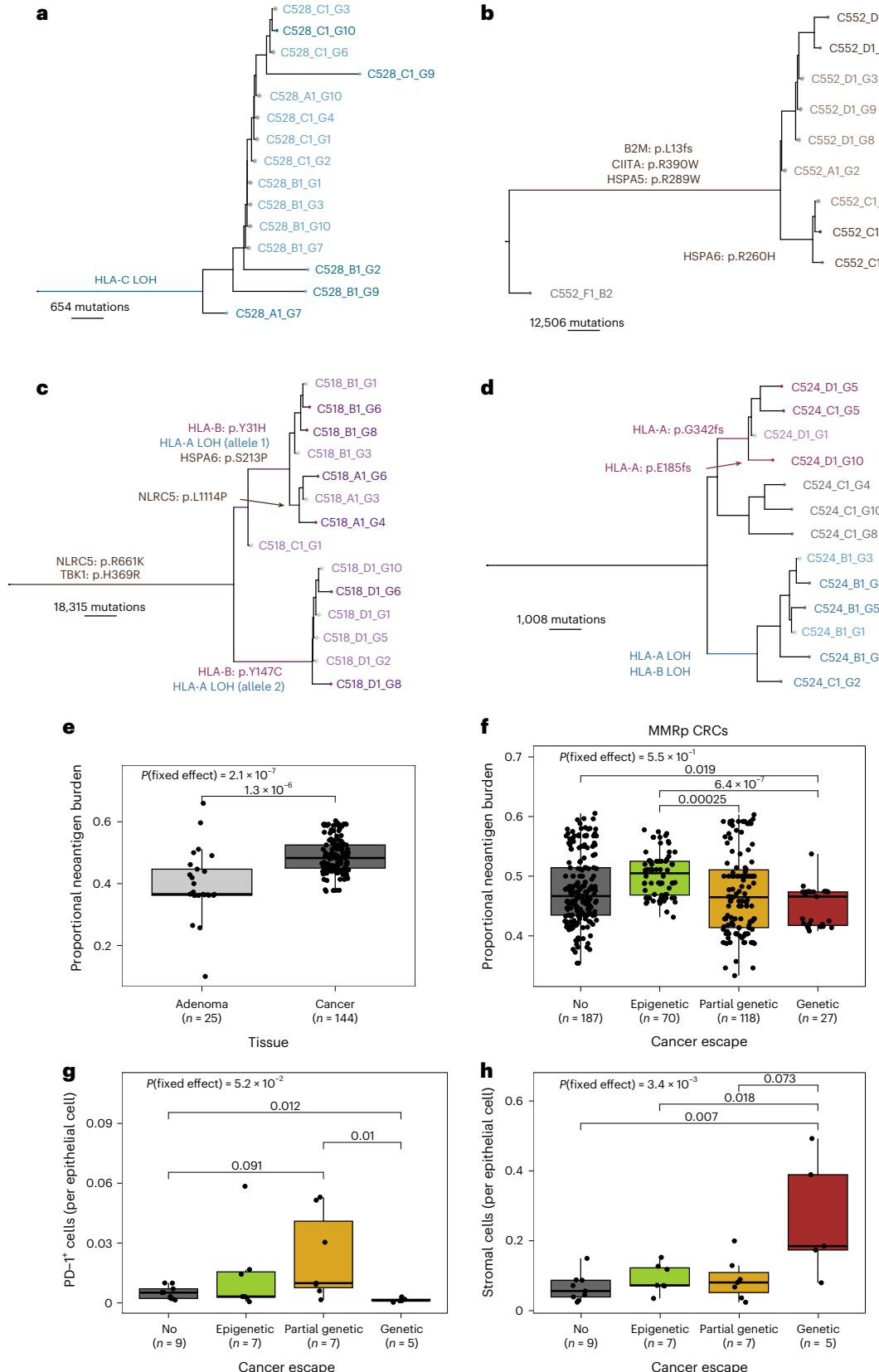

**Fig. 4 | Association between timing of immune escape, editing and exclusion.** **a**–**d**, Phylogenetic tree reconstructed using FF–WGS data of patient C528 (**a**, MMRp), C552 (**b**, MMRd), C518 (**c**, MMRd) and C524 (**d**, MMRp). Immune escape alterations are shown by arrows or over the branch they occurred in, with colors representing mutations as in Fig. 1b. Light-colored samples were sequenced using low-pass WGS and genotyped using deep-sequenced samples. **e**, Proportional burden of CRA and CRC biopsies. **f**, Proportional burden values for MMRp biopsies, according to the immune escape status of each cancer. **g**,**h**, Number of PD-1$^+$ (**g**) and stromal (**h**) cells per epithelial cell in MMRp FFPE–PS samples, according to cancer-immune escape status. Points in **e**–**h** correspond to ROIs/biopsies and the $P$ value of the mixed-effects model incorporating Patient as a random effect is indicated at the top of each panel.

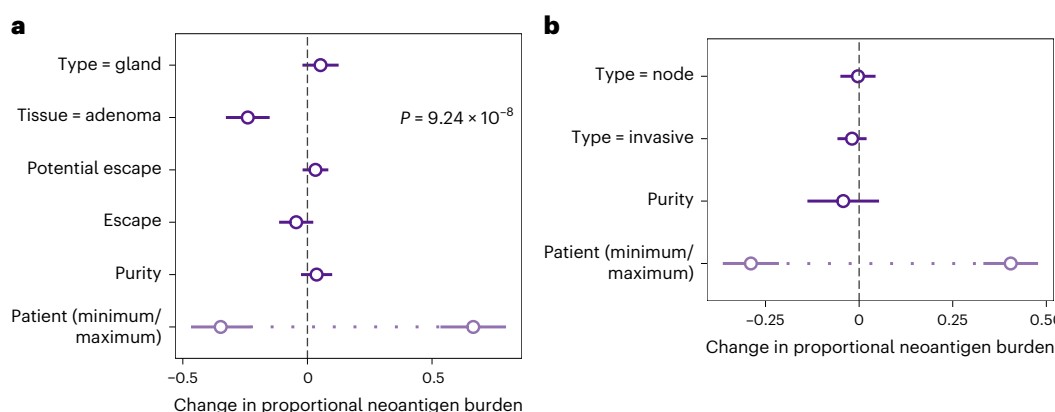

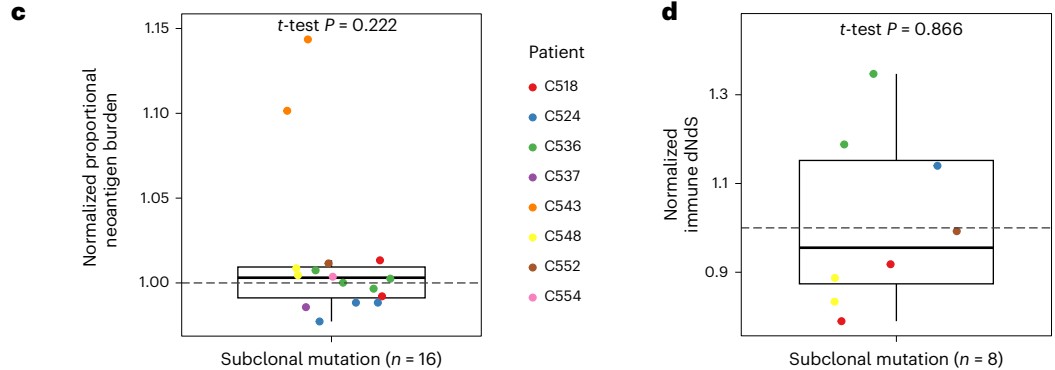

**Fig. 5 | Intra-tumor differences in immune selection. a,b,** Forest plots depicting the output of a multivariable regression (Methods) investigating the association of proportional neoantigen burden with other sample characteristics in FF–WGS (**a**, *n* = 495) and FFPE–PS (**b**, *n* = 82) samples. Circles denote the estimated coefficients, with whiskers showing 95% CIs. A significant *P* value from the multivariable regression is indicated in **a**. Patient-associated coefficients are represented by the range of coefficients estimated, shown in light purple.

**c,d,** Proportional neoantigen burden (**c**) and immune dNdS (**d**) in FF–WGS biopsies with subclonal immune escape. Values have been normalized by the average of all deep-sequenced biopsies phylogenetically close to the escaped biopsy (Extended Data Fig. 1). Points correspond to individual biopsies and cancers are depicted in different colors. *P* values from two-sided one-sample *t*-tests against the null hypothesis of mean = 1 are shown at the top of each panel.

PD-L1+ tumor cells were dispersed within PD-L1+ neighborhoods, even within single tumor glands (Fig. 6c), suggesting high plasticity in PD-L1 expression. We examined colocalisation of CTLs and PD-L1+ cells and found that PD-L1+ epithelial cells were significantly closer to CTLs than PD-L1− epithelial cells, with the closest PD-L1+−CTL relationship observed in the invasive margin (Fig. 6d and Extended Data Fig. 10e,f). Nonepithelial cells expressing PD-L1 were also enriched in the invasive margin, as well as in node deposits (Extended Data Fig. 10g). Moreover, PD-L1+ cells were mixed uniformly with (cognate) PD-1+ cells in the invasive margin and node, whereas they often showed exclusion in superficial tumor and normal tissue (Fig. 6e). These observations suggest that CTL surveillance is heightened at the invasive edge and PD-L1 expression may arise as an adaptive response to it.

In matched genomic data, we explored immune selection signal within the invasive margin samples specifically. We observed that these samples contained candidate highly immunogenic neoantigens at lower VAFs than nonimmunogenic mutations (Kolmogorov–Smirnov test $P = 6 \times 10^{-4}$; Supplementary Methods and Supplementary Fig. 2a). Depletion of neoantigens was most prevalent within small subclones (0.05 < VAF < 0.1, $P = 2 \times 10^{-4}$; Fig. 6f). On the other hand, no neo-antigen VAF depletion was observed in superficial tumor or node deposits (Supplementary Fig. 2b,c), or for low-antigenicity neo-antigens (Supplementary Fig. 2d), in agreement with our earlier finding that sample location was not associated with systematic dif-ferences neoantigen elimination. Therefore, it appears that enhanced immuno-editing occurs as a variable ongoing process at the point

of invasion, affecting primarily very small subclones (~100 cells) of increased immunogenicity.

## Discussion

In this work, we mapped the interplay of epigenetic, genetic and micro-environmental processes that collectively establish immune evasion and shape tumor-immune coevolution in CRC. Our analyses indicate that immune evasion processes are part of the 'Big Bang' that forms CRCs[39], which then specifies the immunogenicity of the whole cancer.

We found that SCAAs contribute to downregulation of neoantigen-carrying genes and genes associated with antigen presentation. We identified several TFs as potential regulators of APGs through SCAA losses. Notably, NFIC binding sites covered all SCAA-affected APGs, although its functional validation as an experi-mental target for immunomodulation in CRC is necessary. Promis-ingly, NFIC has recently been indicated in immunotherapy response in a subgroup of lung cancer patients, where its silencing led to an immune escaped phenotype[40].

We found that immunogenic alterations are depleted preferen-tially in transcription through (so far uncharacterized) epigenetic mechanisms, especially relevant for the control of clonal neoantigens that immune selection probably acts upon[41,42]. Our results align with previous observations that epigenetic and environmental factors play a substantial role in determining cancer prognosis, immune sensitivity and response to immunotherapy[20,43–45]. Consequently, mapping the epigenome could identify patients with lower/higher potential for

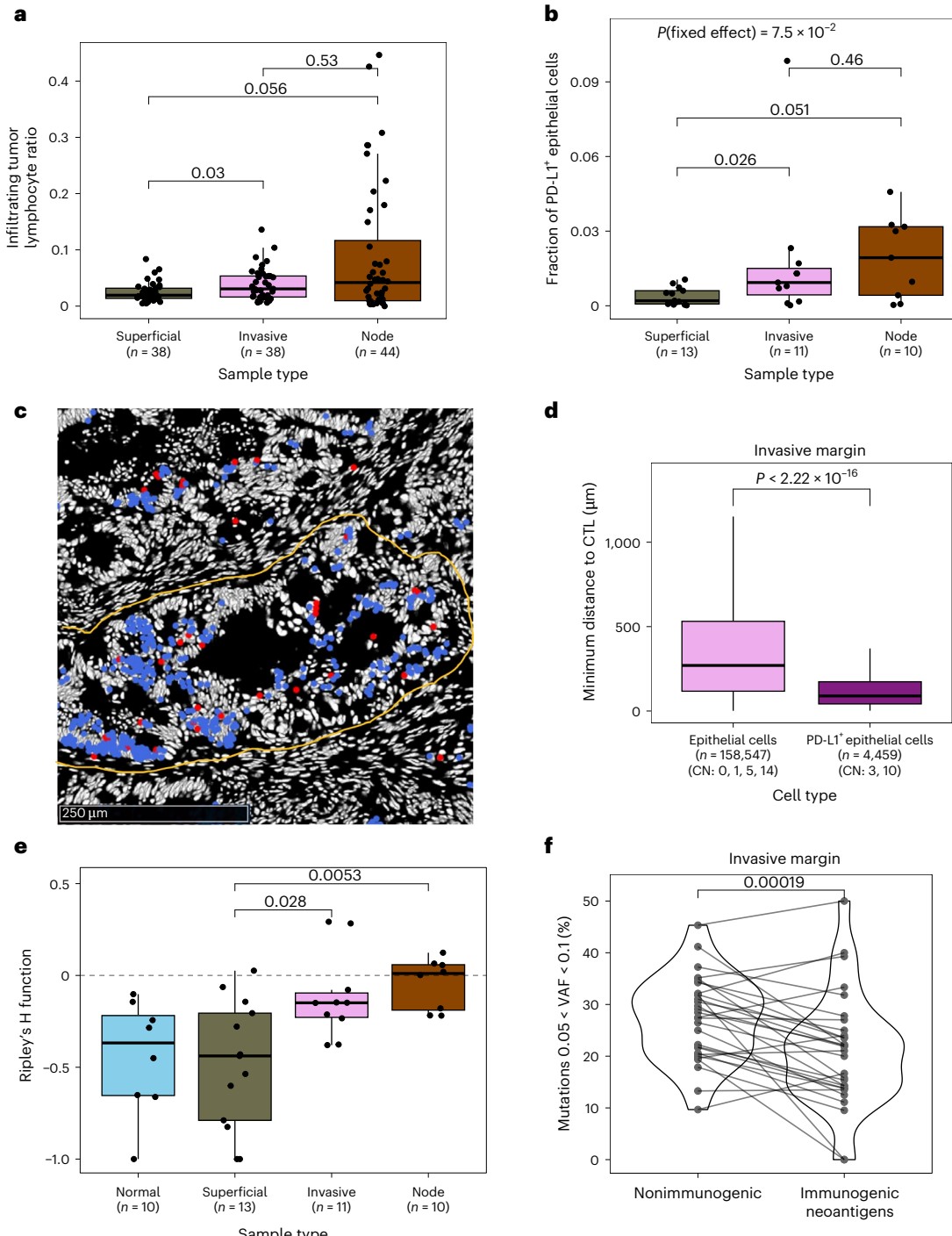

**Fig. 6 | Increased immune interaction and immuno-editing in the invasive margin. a**, Infiltrating tumor lymphocyte ratio quantified using H&E stained images, according to the type of tumor-associated ROI. **b**, The fraction of PD-L1+ epithelial cells in CyCIF images across different sample types. **c**, Representative CyCIF image of a tumor gland from the invasive margin, showing PD-L1+ epithelial cells in blue and cytotoxic T cells in red. The outline of the gland is shown by a yellow line. **d**, Distance of epithelial cells to the closest cytotoxic T cell, for non-PD-L1-expressing epithelial cells in non-PD-L1 CNs and for PD-L1+ epithelial cells in PD-L1-associated CNs ($n = 158,547$ and $4,459$ for non-PD-L1 and PD-L1+

cells, respectively). **e**, Ripley's H function measuring intermixing between PD-L1+ and PD-1+ cells across different cancer regions. Values around zero (dashed line) indicate uniform mixing. **f**, The proportion of all high-immunogenicity neoantigens and of all nonimmunogenic mutations falling within the range $0.05 < VAF < 0.1$ in $n = 29$ invasive margin samples. Each line indicates paired values derived from the same biopsy, with two-sided paired Wilcoxon test $P$ value shown at the top of the panel. Points in **a**, **b** and **e** correspond to ROIs/biopsies and $P$ value of the mixed-effects model incorporating Patient as a random effect is indicated at the top of **b**.

immunotherapy response; large cohort studies are required. Feasibly, drugs modifying the epigenome or selected TF activity could have a profound impact on neoantigen presentation and potentially synergize with immunotherapeutic drugs. This will require future work

on epigenetic target validation in patient-derived co-culture model systems such as organoids[46].

Although there is heterogeneity in the neoantigen burden and composition of the immune microenvironment, we observed that

clonal features are most important for determining immunogenicity. We found that the acquisition of immune tolerance happens before or during early carcinoma expansion, as microenvironmental remodeling was prevalent throughout all tumor-associated regions, coupled with high-impact immune escape genetic alterations and epigenetic processes to ensure reduced neoantigen presentation and immune activation. This results in a 'Big Bang' tumor-immune coevolution where pre-expansion somatic alterations and TME features determine the clonal neoantigen burden of these cancers and set the trajectory of evolution, with ongoing (subclonal) selection forces playing only a weak role. This is in agreement with our previous work that also highlighted a shift in immunogenicity before carcinoma expansion[14]. Collectively, these data suggest that CRC are no longer engaged in an 'all-out war' with the immune system after initial expansion. We found that ongoing 'battles' of cancer-immune interaction are limited to small localized subclones along the invasive margin of MMRp CRCs, where we presume that there is a large change in immune microenvironmental composition and hence more pronounced immuno-editing. These subclones are characterized by an intricate interplay of higher CTL infiltration, depletion of subclonal neoantigens and PD-L1+ expression, painting the picture of an 'ongoing skirmish' between active immune and cancer cells. However, we did not observe any variation associated with subclonal immune alterations within the superficial tumor, confirming that the tumor bulk is not shaped by such ongoing battles.

Overall, our work indicates the acquisition of immune escape and/ or an immune excluded microenvironment is part of the 'Big Bang' necessary for early carcinoma expansion in CRCs. We observe that epigenetic, transcriptomic and microenvironmental mechanisms add a layer of regulation to the immunogenomic profile of CRC cells, allowing further adaptation through both hereditary and plastic means. Our findings suggest evaluating and targeting of epigenetic machinery and TME reorganization as a new way to stratify patients and potentially enhance the efficacy of immunotherapy.

## Online content

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

[1]Centre for Evolution and Cancer, The Institute of Cancer Research, London, UK. [2]Department of Mathematical Sciences, Chalmers University of Technology and University of Gothenburg, Gothenburg, Sweden. [3]UCL Cancer Institute, University College London, London, UK. [4]Centre for Genomics and Computational Biology, Barts Cancer Institute, Queen Mary University of London, London, UK. [5]Computational Biology Research Centre, Human Technopole, Milan, Italy. [6]Arizona Cancer Evolution Center, Biodesign Institute and School of Life Sciences Arizona State University, Tempe, AZ, USA. [7]St. Anna Children's Cancer Research Institute, Vienna, Austria. [8]Histopathology Department, University College London Hospitals NHS Foundation Trust, London, UK. [9]These authors contributed equally: Eszter Lakatos, Vinaya Gunasri. [10]These authors jointly supervised this work: Andrea Sottoriva, Trevor A. Graham. ✉e-mail: eszter.lakatos@chalmers.se; andrea.sottoriva@fht.org; trevor.graham@icr.ac.uk

## Methods

### Sample collection and sequencing

Our FF–WGS samples were comprised of processed data from previous sequencing experiments of our evolutionary predictions in CRC cohort, which has been described in refs. 15,23. All patients gave informed consent in writing for collection of their materials to the UCLH Cancer Biobank (Research Ethics Committee approval 15/YH/0311). All investigators were blinded to patient data related to outcome, and all clinicopathological information.

Our FFPE–PS samples originated from 11 stage III microsatellite-stable cancers with lymph node metastases from the same cohort, 9 of which are shared with FF–WGS samples. FFPE sections were cut in the following order: H&E-1 (5 µm), 5 × 8 µm for laser capture microdissection, H&E-2 (5 µm), 8 × 5 µm for CyCIF, H&E-3 (5 µm). The H&E slides from each FFPE block were digitized using the NanoZoomer S210 or S60 (Hamamatsu). Images were reviewed using NDPViewer software (v.2.9.29).

Regions of interest (ROIs) were identified as single glands or clusters of small adjacent glands (microbiopsies) from superficial, invasive margin or lymph node deposits. Superficial regions were defined as cancer regions adjacent to or contiguous with normal mucosa. The tumor–normal interface was identified where possible, and the invasive margin was defined as the region within 500 µm either side of the tumor–normal interface (with an overall extent of ~1 mm).

**Panel sequencing.** ROIs were microdissected using PALM MicroBeam Laser Microdissection (Zeiss). DNA was extracted using the High Pure FFPET DNA Isolation Kit (Roche). Extracted FFPE DNA was repaired using the NEBNext FFPE DNA Repair Mix. Postrepair, whole-genome libraries were prepared using the NEBNext Ultra II FS DNA Library Prep Kit for Illumina with unique molecular identifier (UMI) adapters ligated onto DNA molecules.

PS on FFPE samples was carried out using a custom targeted panel designed by L. Zapata and manufactured by Twist BioSciences, focusing on regions encoding the immunopeptidome and antigen-presenting related genes. The immunopeptidome was defined as the set of the human nine-mers that bind strongly to one of the top 70 *HLA* alleles, was confirmed T cell positive in IEDB and were derived from a gene with mean expression of more than one fragments per kilobase million pancancer. The final list of these immunopeptidome loci can be obtained from: https://github.com/luisgls/SOPRANO/blob/master/immunopeptidomes/human/allhlaBinders_exprmean1.IEDBpeps.unique.bed.

The Twist Target Enrichment Protocol was used to hybridize probes from the this custom panel with prepared libraries. Hybridized targets were isolated and amplified with PCR. Paired-end 50-bp runs were performed on Novaseq S1 (Illumina).

### Processing of sequencing data

The processing of FF–WGS data was detailed in ref. 15.

For FFPE–PS samples, three sets of fastq reads were aligned to human reference genome build hg38 to generate an unmapped bam using FastqToBam (Fgbio v.1.3.0). Fastq files were then created from this unmapped bam (SamToFastq, Picard v.2.20.3) and aligned to human reference genome build GRCh38/hg38 with Burrows–Wheeler Aligner package BWA-MEM v.0.7.17. Alignment data from the outputted bam from this step were merged with data in the previously generated unmapped bam (using the MergeBamAlignment function of Picard v.2.20.3). The merged bam file was then used as input for GroupReadsByUMI (Fgbio v.1.3.0). 'Adjacency' was used as the strategy for grouping so that errors were allowed between UMIs, but only when there was a count gradient. The allowable number of edits/changes between UMIs was set to 1, and minimum mapping quality to 30 (default).

Consensus sequences were called from reads with the same unique molecular tag (CallMolecularConsensusReads) and filtered using FilterConsensusReads to exclude consensus sequences with fewer than two contributing reads, mask consensus bases with quality less than 30, accept a maximum raw-read error rate across the entire consensus read of 0.05, accept a maximum error rate for a single consensus base of 0.1 and accept a maximum fraction of 0.2 for no-calls in the read after filtering.

The filtered reads were then re-aligned to GRCh38/hg38, and variant calling was performed using Mutect2 (v.4.1.4.1) with a bed file specifying regions of the genome covered by the targeted panel. Normal samples from adjacent muscle were used as matched normal for variant calling. The resulting vcf files were merged for each patient and passed to Platypus (v.0.8.1.1) for multiregion variant calling.

Variant calls were filtered to retain mutations within certain filtering criteria and adequate support for the variant, as described in ref. 15. The same filters were used for both FF–WGS and FFPE–PS samples, except for requiring at least five and at least eight reads covering each site in FF–WGS and FFPE–PS samples, respectively.

FF–WGS samples sequenced at shallow depth were genotyped and fitted on phylogenetic trees using the method described in ref. 15.

Raw RNA-seq read counts were normalized and converted using DESeq2 as described in ref. 15. The code for reproducing RNA-seq processing can be found at https://github.com/JacobHouseham/EPICC_transcriptome. Raw counts were converted either to transcript per million (TPM) values or processed counts following VST, see '2.gene_expression_normalisation_and_filtering.Rmd' within the above repository. TPM values were used to assess expression levels of genes (for example, to establish constitutively expressed genes), while VST values were used for between-sample comparison of a particular gene.

### HLA haplotyping

*HLA-A*, *-B* and *-C* haplotyping was performed using polysolver[47] (v.1.0) on FF–WGS samples, by running shell_call_hla_type with default settings. As ethnicity information was not available, we used 'Unknown' for all samples. To increase coverage, we used merged bam files created from all sequencing files from a given cancer. For validation, we also performed haplotyping on merged bams formed of normal (blood or normal colon tissue) samples and compared the predicted haplotypes. The predicted haplotypes had a high concordance, with haplotypes predicted using all samples providing one more heterozygous haplotype than normal-only haplotypes in 6 of 30 cases. Based on the average homozygosity across CRCs (as seen in TCGA CRC samples[8]), we accepted the more heterozygous set of alleles predicted to define their set of *HLA* alleles. For all *HLA* alterations, we confirmed that the alterations are called independent of which haplotyping calls were used.

For FFPE–PS samples we used the calls derived from matched FF–WGS samples. For the two patients where this was not available, we performed haplotyping on adjacent normal mucosal samples following the same steps as described above.

### Immune escape prediction

**Mutations in HLA.** Somatic mutations in the *HLA* locus were predicted using polysolver[47] (v.1.0). The mutation detection script of polysolver (shell_call_hla_mutations_from_type) was run on matched tumor–normal pairs to call tumor-specific alterations in HLA-aligned sequencing reads using MuTect (v.1.16). In addition, Strelka2[48] (v.2.9.10) was run independently to detect short insertions and deletions in HLA-aligned with increased sensitivity. Both single nucleotide mutations and frameshift alterations passing quality control were annotated by shell_annotate_hla_mutations. Based on this annotation, a mutation in the *HLA* locus was called if a mutation passed all quality filters and introduced either a missense/nonsense change or was located at a splice site. In addition, we also identified second-tier mutations in unfiltered MuTect files that were detected in insufficient reads to pass quality control, but the exact same nucleotide change was clearly detected in another biopsy of the same tumor, allowing detection in samples with low purity or sequencing coverage.

**LOH in HLA.** LOH at the *HLA* locus was predicted using LOHHLA[33] and sequenza[49]. First, we evaluated the allele-specific copy number as predicted by sequenza (v.2.1.2) at the *HLA-A*, *HLA-B* and *HLA-C* loci. Samples with a predicted minor allele copy number of 0 (for example, 2:0, 3:0) were labeled as candidate LOH. Then, we ran LOHHLA with the polysolver-generated haplotype files and matched tumor–normal pairs as input. A type I allele of a patient was annotated as 'allelic imbalance' (AI) if the associated *P* value was lower than 0.01. Alleles with AI were labeled as LOH if the following criteria held: (1) the predicted copy number of the lost allele was below 0.5 with confidence interval (CI) strictly below 0.7; (2) the copy number of the kept allele was above 0.75; (3) the number of mismatched sites between alleles was above 10.

HLA LOH was identified by both methods independently in 18 deep-sequenced samples. HLA LOHs called by only one of the methods were inspected manually. Of LOHHLA-exclusive calls, two were found to be false positives and four were identified as AIs with a minor allele copy number of 1. Of sequenza-exclusive calls, one region showed clear LOH and got classified as HLA LOH; six samples showed similar CN pattern but were missed by LOHHLA due to low purity—as adjacent low-pass sequenced samples showed CN = 1 in the *HLA* region, we also classified these as HLA LOH. In addition, 25 regions had allelic imbalance detected by LOHHLA that were also confirmed in sequenza to have a CN state of 2:1 or 3:1.

**Mutations in APGs.** We assembled a list of genes involved in class I type MHC presentation, by using the KEGG pathway 'antigen processing and presentation' and MHC I pathway specifically. The following genes were considered: *TAP1*, *TAP2*, *IRF1*, *NLRC5*, *TBK1*, *PSME3*, *PSME1*, *ERAP2*, *ERAP1*, *HSPBP1*, *CALR*, *B2M*, *PSME2*, *PSMA7*, *CANX*, *CIITA*, *TAPBP*, *CREB1*, *HLA-A*, *HLA-B*, *HLA-C*, *HSP90AA1*, *HSP90AB1*, *HSPA2*, *HSPA4*, *HSPA5*, *HSPA6*, *HSPA8*, *IFNG*, *NFYA*, *NFYB*, *NFYC*, *RFX5*, *RFXANK* and *RFXAP*. Then, we evaluated the expression of each gene within our cohort and filtered out genes that were not clearly expressed (≥10 TPM) in at least 5% of samples.

Then, we called evaluated the mutations called in these genes following. Only mutations with at least moderate predicted impact were called.

**Neoantigen prediction and proportional burden computation**
We predicted neoantigens from somatic mutation calls and patient-specific HLA haplotypes using NeoPredPipe[26] for both FF–WGS and FFPE–PS samples. We defined neoantigen burden in a sample as the number of (unique) mutations giving rise to at least one strong-binding (rank <0.5) neoantigen. We focused our analysis on SNV neoantigen burden, unless stated otherwise. We also computed the total protein-changing mutation burden, and used this value to obtain the proportional neoantigen burden for each sample, that is, what percentage of mutations that has the potential to create a neoantigen does actually lead to strong-binder neoantigens.

**Defining clonal/subclonal neoantigens.** We assigned clonal/subclonal categories to all mutations (independent of neoantigen status) based on their presence/absence in all available deep- or panel-sequenced sample of a given cancer. As the targeted genome region, sequencing strategy and sample types were different, we created separate mutations lists of FF–WGS and FFPE–PS samples. For FF–WGS samples, mutations present in all sequenced cancer biopsies were denoted as clonal and all other mutations (absent in at least one biopsy) as subclonal. For FFPE–PS samples, mutations present in all biopsies were deemed clonal, and mutations absent in at least two biopsies were denoted as subclonal.

**Immune dNdS**
Immune dNdS was computed using SOPRANO[25], with somatic mutation files and personalized immunopeptidome files derived specifically to HLA haplotypes. First, the ratio between dNdS inside (ON-target dNdS) and outside the immunopeptidome (OFF-target dNdS) was computed and corrected for 192-trinucleotide context. Then immune dNdS was computed as the ratio of ON-to-OFF dNdS to correct for technical artefacts that could bias dNdS as computed in OFF-target regions. Samples without any ON- or OFF-target synonymous mutations were excluded from the analysis. In total, immune dNdS estimate was available for 61 FF–WGS and 41 FFPE–PS samples.

To compute immune dNdS separately, we first filtered somatic mutation files to only contain mutations annotated as clonal, then repeated the above procedure on these files.

**SCAA and SCAA loss analysis**
We identified SCAAs by comparing purity-corrected and copy number-corrected ATAC-seq peak calls of cancer regions (per cancer) to a pool of normal glands[15]. For immune escape genes, we filtered SCAAs for those located in promoter or enhancer regions associated with a gene from the list detailed in 'Mutations in antigen-presenting associated genes.' Each SCAA was labeled as loss (fold change of cancer compared to normal < −1) or gain (fold change of cancer compared to normal > 1).

To evaluate the observed number of SCAA losses, we repeated the analysis 200 times with a set of 25–40 randomly chosen genes and computed the number of SCAA losses and gains. We derived a *P* value as the number of random samples we observed a value more extreme than for APGs.

For neoantigen SCAA loss analysis, we filtered SCAA calls to obtain only losses located in promoter regions (within 1,000 bp of the transcription start site of a gene). Each SCAA loss was annotated by the gene to which they were proximal. As most SCA alterations were found to be clonal, we defined SCAA losses on the cancer level.

We then evaluated for each protein-alteration mutation within a cancer whether the gene it is in had an associated SCAA loss or not. For each cancer, we computed the proportion of mutations (both for neoantigens and for nonantigenic mutations) that were classified as downstream of a SCAA loss. Similarly, for each SCAA loss in a given cancer, we evaluated whether it was upstream of a protein-changing mutation and, if so, within each cancer, we counted the number of SCAA losses upstream of neoantigen/nonantigenic mutations.

**TF analysis.** We used the TF binding sites obtained from curated human TF motifs in ref. 21, also available as part of the processed Mendeley dataset ('Data availability').

We examined the list of ATAC-seq peaks that showed statistically significant somatic loss in at least one cancer (27 peaks, representing 20 unique APGs). We then filtered this set to peaks with loss in more than one patient (recurrent SCAA losses), leading to a total of ten genomic regions associated with eight APGs. We identified TFs that are predicted to bind to these regions as those with a binding site in <1,000 bp distance of a given peak, using the distanceToNearest function of the GenomicRanges package in R. We selected for plotting ten TFs that bound more than two of the regions (Extended Data Fig. 2a).

**Transcriptional editing analysis**
For each SNV detected in FF–WGS samples, we quantified the number of reads in the matched RNA-seq data supporting the mutation using bam-readcount (v.1.0.1)[50]. For a mutation to be classified as expressed in a sample, we required three or more RNA reads overlapping the position to support the variant base. For a mutation to be classified as not expressed (that is transcriptionally edited), we required more than ten overlapping reads, with zero supporting the variant. We chose the threshold of more than ten so that the probability of misclassifying a mutation present at a true allele frequency of 0.25 or higher was <5%. Mutation–sample pairs that did not qualify for either of these categories were left blank to signify insufficient evidence.

For each cancer, we computed the proportion of mutations that had evidence of transcriptional editing (present in WGS but not expressed in RNA-seq) in at least one sample. Similarly, for each mutation in a cancer, we computed the number of samples that expressed that mutation and the number of samples that (confidently) did not express it. We excluded mutations that had sufficient evidence for expressed/not status in fewer than two samples. We used the median proportion of biopsies with evidence of editing to derive a single value per cancer per mutation type.

### Analysis of H&E slides

Sections were cut from diagnostic FFPE blocks sampled at resection. H&E slides pre- and post-laser capture microdissection (LCM) and post-CyCIF were scanned using the NanoZoomer S210 slide scanner (Hamamatsu). Representative sections were selected for annotations, which were drawn on pre-LCM H&E slides as first choice in most cases. Annotations included: all tumor (on a slide); the tumor–normal interface; any deposits of cancer within nodes; and adjacent normal mucosa (within 5 mm of superficial tumor) and normal mucosa further away (>5 mm away from tumor).

Annotations were made using the NDPViewer software (v.2.9.29). The tumor–normal interface was expanded by 500 μm on either side to identify the invasive margin. Images with annotations were analyzed using a digital cell classifier, which uses deep learning methodology modeled on a spatially constrained neural network architecture[51]. The model first detects cells through predicted location of cell nuclei, then classifies them as: normal epithelial, cancer epithelial, fibroblast, lymphocyte, neutrophil, macrophage, endothelial. Absolute counts are calculated for each annotated region. For each annotation, number of cells were normalized to the total number of epithelial cells to obtain per epithelial cell counts.

In addition, we used a further classifier[37] to class all tumor-associated lymphocytes as infiltrating, adjacent or distant based on proximity to tumor cells.

### CyCIF imaging

ROIs that matched the cancer glands/microbiopsies used for LCM were identified using the H&Es pre-LCM and post-LCM and extended by an additional 1 mm at the edges. All CyCIF sections were 5 μm thick. The maximum separation between the LCM slides and the CyCIF slide was 10 μm.

The protocol was based on previous methods from ref. 24. Full details of fluorescent antibodies are listed in Supplementary Table 2. Image and statistical analysis of CyCIF images is detailed in Supplementary Methods.

### Statistical analysis

All statistical analyses were carried out in R (v.4.4.2). All comparisons between pairs of samples were carried out using two-sided unpaired Wilcoxon rank sum test, without additional adjustment of $P$ values, unless stated otherwise. In all figures, visual elements of the boxplots correspond to the following summary statistics: center line, median; box limits, upper and lower quartiles; whiskers, 1.5× interquartile range.

**Mixed-effect model with patient effect.** To account for multiple samples originating from the same patient, we used a mixed-effects model implemented with the R package lme4 (v.1.1.36). We incorporated Patient as a random effect and the tested variable (for example CRA/CRC) as a fixed effect affecting the intercept into the mixed-effects model. The significance of the fixed effect was evaluated by comparing this 'full' model to one with only random effects, using ANOVA. The $P$ value of this test (Pr(>Chisq)) is reported as $P$(fixed effect).

**Evaluating neoantigens in expressed genes and transcriptionally/ epigenetically silenced neoantigens.** We created 2 × 2 contingency tables by counting mutations that are neoantigens/nonantigenic and (1) in a gene in the given gene group or not; (2) in a consistently expressed gene or not; (3) in a gene with somatically closed promoter or not or (4) not expressed (missing) in at least one sample or not. Fisher's exact test was used to compute the OR and CIs on these tables. The above steps were repeated separately on clonal and subclonal mutations alone.

**Multivariable regression.** Multivariable regression models were constructed using the functions betareg (for proportional burden) and lm (for immune dNdS). Sample type (gland/bulk for FF–WGS and superficial tumor/IM/node for FFPE–PS), tissue type (adenoma/cancer), immune escape (escape/weak-escape/no-escape) and patient were encoded as categorical variables; purity as a continuous variable. For Extended Data Fig. 10, sample type and patient were encoded as categorical variables and all immune infiltrate values handled as continuous variables (unit: number per epithelial cell). Results were visualized using the plot_summs function from the package jtools, omitting the full list of coefficients assigned to patients (as well as sample type for Extended Data Fig. 10a) to make the visuals easier to interpret. Instead, on Fig. 5a,b, the smallest and largest patient-associated coefficients were identified and plotted to highlight the range of coefficients.

### Reporting summary

Further information on research design is available in the Nature Portfolio Reporting Summary linked to this article.

### Data availability

Processed data used in Figs. 1–6, Extended Data Figs. 1–10 and Supplementary Figs. 1–2 and to derive summary tables are available at Mendeley: https://doi.org/10.17632/cjfmmc95dm.2. Raw sequencing reads of FFPE–PS samples are available on the European Genome-Phenome Archive at accession code EGAS50000001154. Raw sequencing data of FF–WGS samples is available at accession code EGAS00001005230, and previously generated processed data at https://data.mendeley.com/datasets/7wx3chtsxx/2.

### Code availability

Scripts required to reproduce data processing and Figs. 1–6, Extended Data Figs. 1–10 and Supplementary Figs. 1–2 are available via GitHub and Zenodo at https://github.com/elakatos/EPICC_immune_analysis and https://doi.org/10.5281/zenodo.16599726 (ref. 52).

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

### Acknowledgements

This work was supported by the Wellcome Trust (202778/Z/16/Z to T.A.G.; 202778/B/16/Z to A.S.; 105104/Z/14/Z to the Center for Evolution and Cancer, Institute of Cancer Research) and Cancer

Research UK (A19771 and DRCNPG-May21_100001 to T.A.G. partially supporting E.L.; A22909 to A.S.; Clinical Research Training Fellowship supporting V.G., Accelerator Award A26815 supporting A.S. and latterly the CRC STARs strategic award SEBCRCS-2024/100001 to T.A.G.). E.L. is supported by grants from Chalmers Area of Advance Health Engineering, Gender Initiative for Excellence, the Swedish Research Council (VR2024-04145 to E.L.) and Adlerbertska Stiftelsen. V.G. received support through a Clinical Lecturer Grant from the Jean Shanks/Pathological Society (JSPS/CLG/1022/01 to V.G.). This work was also supported in part by National Institutes of Health (NIH) grants U54 CA217376, U2C CA233254 and R01 CA140657 to C.C.M. The findings, opinions and recommendations expressed here are those of the authors and not necessarily those of the universities where the research was performed or the NIH.

## Author contributions

A.S. and T.A.G. conceived the study and A.S., T.A.G., E.L. and V.G. designed research. C.C.M., A.S. and T.A.G. acquired funding for the project. E.L., V.G. and T.A.G. wrote the original manuscript with contribution from A.-M.B. M. Mossner, I.S., C.K., M. Mitchison, M.J., M.R.-J. and J.B. contributed to sample collection and data generation. V.G. performed FFPE–PS sequencing and CyCIF analysis with support from A.-M.B. E.L. performed multiomic FF–WGS analysis. V.G. and E.L. performed FFPE–PS bioinformatic analysis. L.Z., J.H., T.H., C.L., G.D.C. and G.L.-P. contributed to bioinformatic analyses. N.T., O.S. and L.C. contributed to image analysis. All authors reviewed and approved the manuscript.

## Competing interests

T.A.G. and A.-M.B. are listed as coinventors on patent application GB2305655.9 that concerns T cell receptor sequencing of cancers, and T.A.G. is a coinventor on patent application GB2317139.0 that concerns measurement of cancer evolutionary dynamics and GB2501439.0 that concerns evolutionary dynamics of cell lines under treatment. T.A.G. has received an honorarium from Genentech Inc. and provides consultancy for DAiNA therapeutics. The other authors declare no competing interests.

## Additional information

**Extended data** is available for this paper at https://doi.org/10.1038/s41588-025-02349-1.

**Correspondence and requests for materials** should be addressed to Eszter Lakatos, Andrea Sottoriva or Trevor A. Graham.

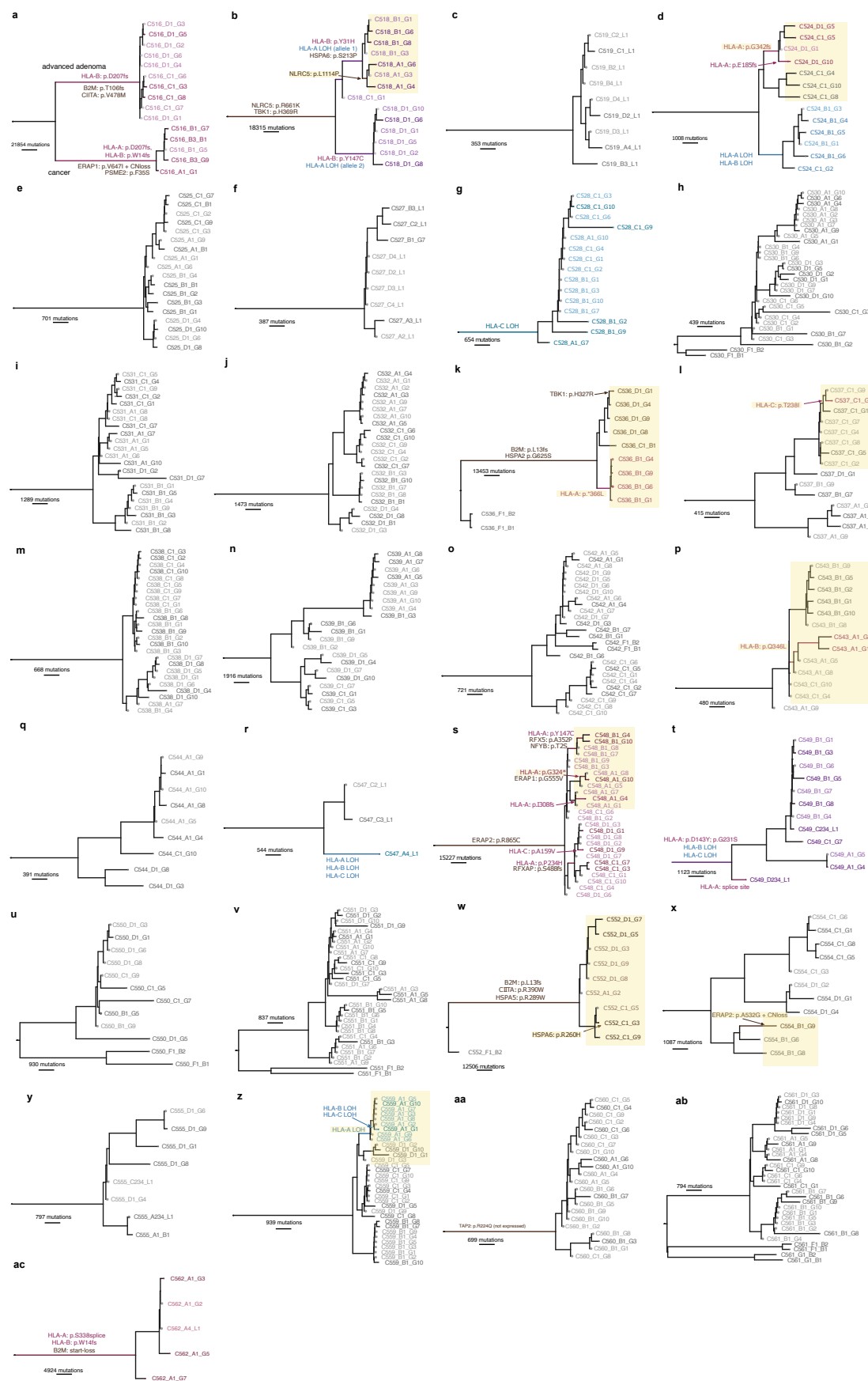

**Extended Data Fig. 1 | See next page for caption.**

**Extended Data Fig. 1 | Phylogenetic trees reconstructed using FF-WGS data of the 29 CRCs.** (**a**–**ac**) Tree constructed for cancer C516 (**a**), C518 (**b**), C519 (**c**), C524 (**d**), C525 (**e**), C527 (**f**), C528 (**g**), C530 (**h**), C531 (**i**), C532 (**j**), C536 (**k**), C537 (**l**), C538 (**m**), C539 (**n**), C542 (**o**), C543 (**p**), C544 (**q**), C547 (**r**), C548 (**s**), C549 (**t**), C550 (**u**), C551 (**v**), C552 (**w**), C554 (**x**), C555 (**y**), C559 (**z**), C560 (**aa**), C561 (**ab**), C562 (**ac**).

Immune escape alterations are shown by arrows or over the branch they occurred in. Light-coloured samples were sequenced using low-pass WGS and genotyped using deep-sequenced samples. Biopsies in yellow shaded areas in (b,d,k,l,p,s,w,x & z) are subclones considered for subclone-specific normalised burden and immune dNdS analysis in Fig. 5c, d.

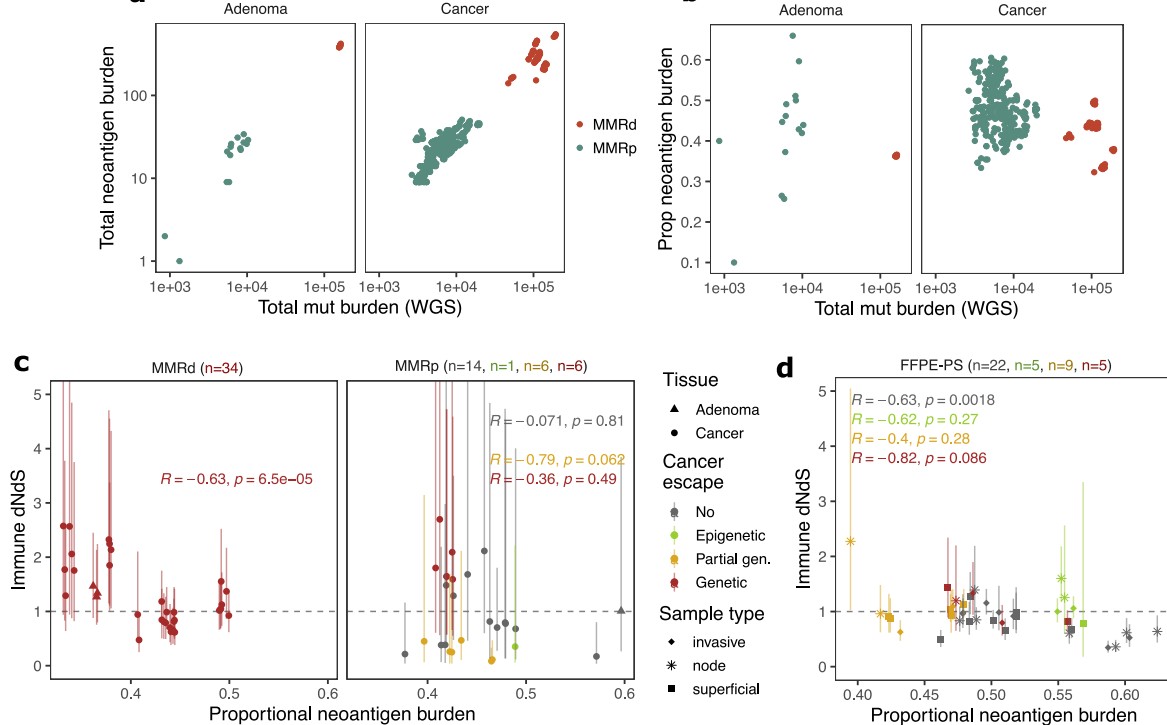

**Extended Data Fig. 2 | Association between mutation burden and immunogenicity measures. (a, b)** Total (**a**) and proportional (**b**) neoantigen burden plotted against total mutation burden in all MMRp (green) and MMRd (red) colorectal adenomas and cancers. (**c, d**) Proportional burden and immune dNdS plotted for fresh frozen whole genome sequencing (FF-WGS) (**c**) and formalin-fixed paraffin-embedded panel sequencing (FFPE-PS) (**d**) samples. Symbols depict immune-dNdS estimates with error bars showing the corresponding confidence intervals, truncated at 5 for FF-WGS samples.

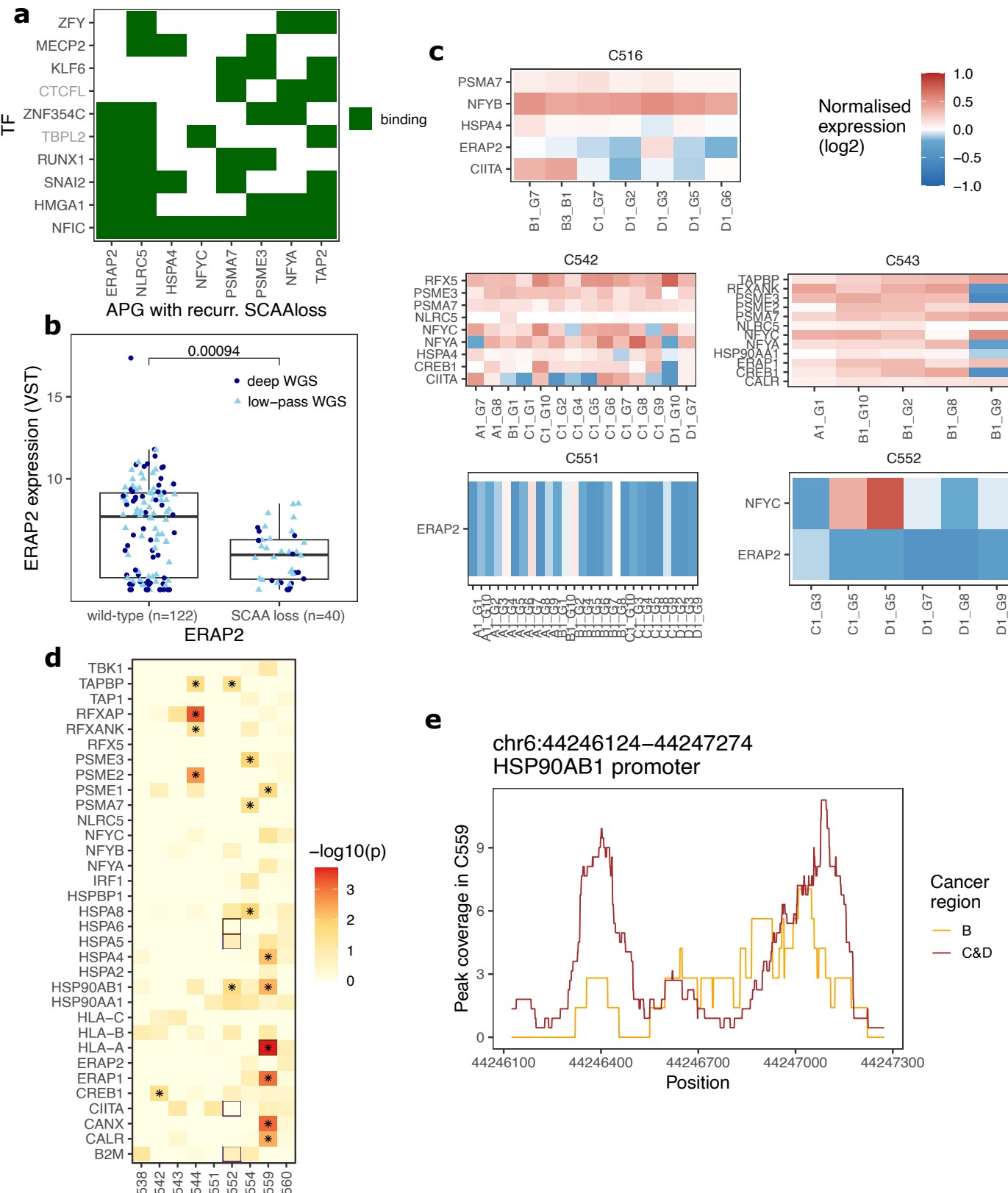

**Extended Data Fig. 3 | Chromatin-associated and heritable regulation of antigen presenting genes.** (**a**) Transcription factors (TFs) that bind >2 of the promoter regions of recurrently SCAAloss-affected APGs. Grey colour denotes TFs that are not confirmed to be expressed in RNAseq. (**b**) Expression of ERAP2 in samples with and without a SCAAloss in the promoter of the gene. (**c**) Normalised expression (compared to non-tumour samples) of antigen presenting genes affected by SCAAloss. (**d**) Phylogenetic signal of the expression of genes associated with immune escape. Cancers where expression of a particular gene is phylogenetic (p < 0.05) are indicated by asterisks. Cancer-gene combinations with a somatic mutation of the gene are indicated by brown rectangles. (**e**) Peak coverage of the HSP90AB1 promoter region in cancer C559. Maroon and orange lines show the average across all samples within region C&D and B, respectively.

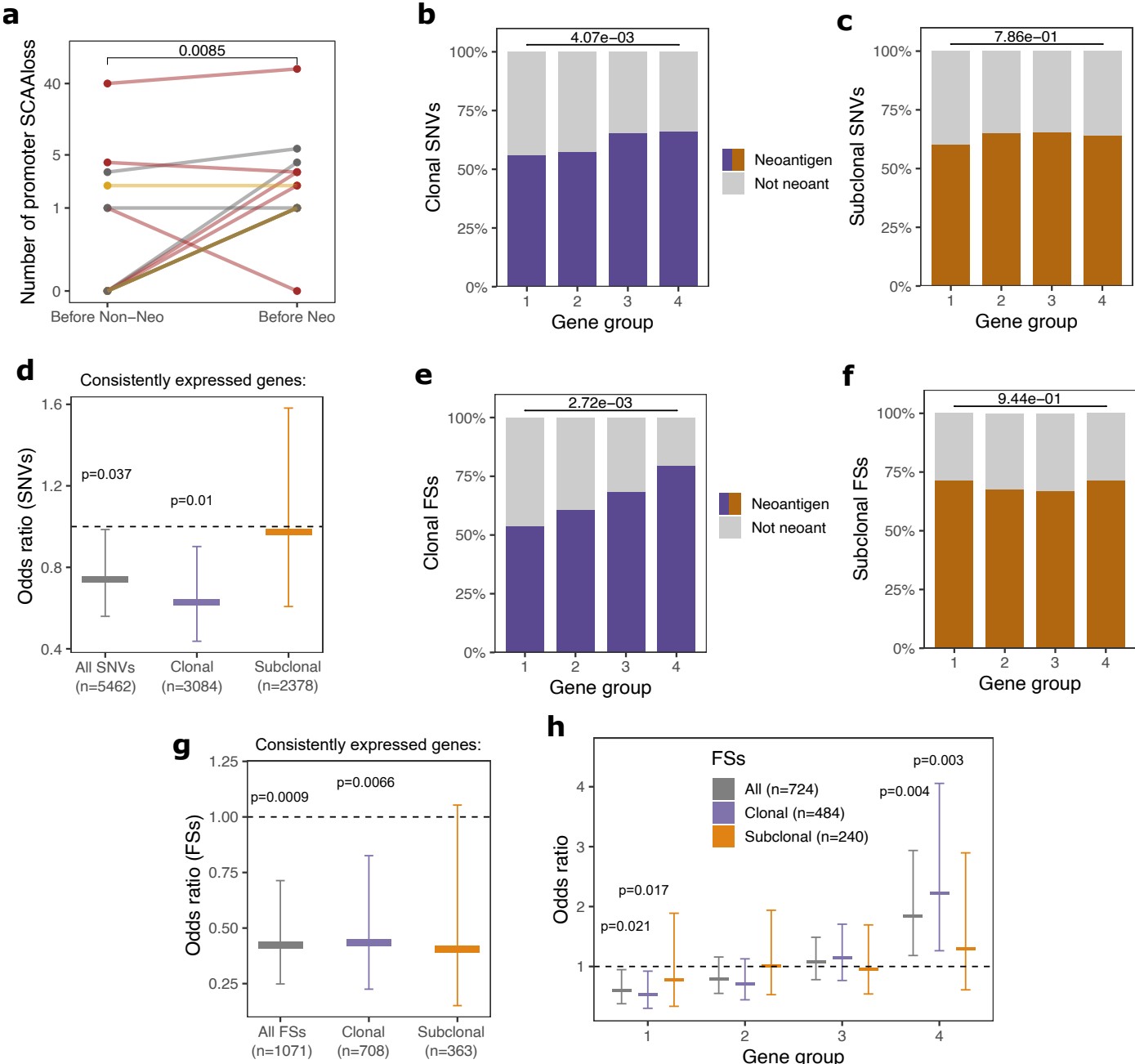

**Extended Data Fig. 4 | Epigenetic and transcriptomic immuno-editing.**
(**a**) Total number of SCAAlosses falling in promoters of genes with an antigenic or non-antigenic mutation in n = 15 cancers. p-value from a two-sided paired Wilcoxon-test is indicated on top of the panel. (**b, c,e, f**) The proportion of antigenic (in colour) and non-antigenic (in grey) single nucleotide variants (SNVs) (**b, c**) and frameshift mutations (FSs) (**e, f**) that are located in genes of gene group 1–4 from ref. 31. Clonal (**b, e**) and subclonal (**c, f**) mutation are shown separately. The p-value of a two-sided chi-squared test comparing groups is shown on top of each panel. (**d, g**) Odds ratio of a neoantigen vs non-antigenic SNVs (**d**) and FSs (**g**) being located in consistently expressed genes. (**h**) Odds ratio of a neoantigen vs non-antigenic FS mutation being located in genes of gene group 1–4 from ref. 31. All, clonal and subclonal mutations are shown separately in grey, purple and orange. In d,g-h markers represent OR values, error bars show corresponding 95% confidence intervals, and significant p-values from unadjusted two-sided Fisher's exact tests are indicated above corresponding bars.

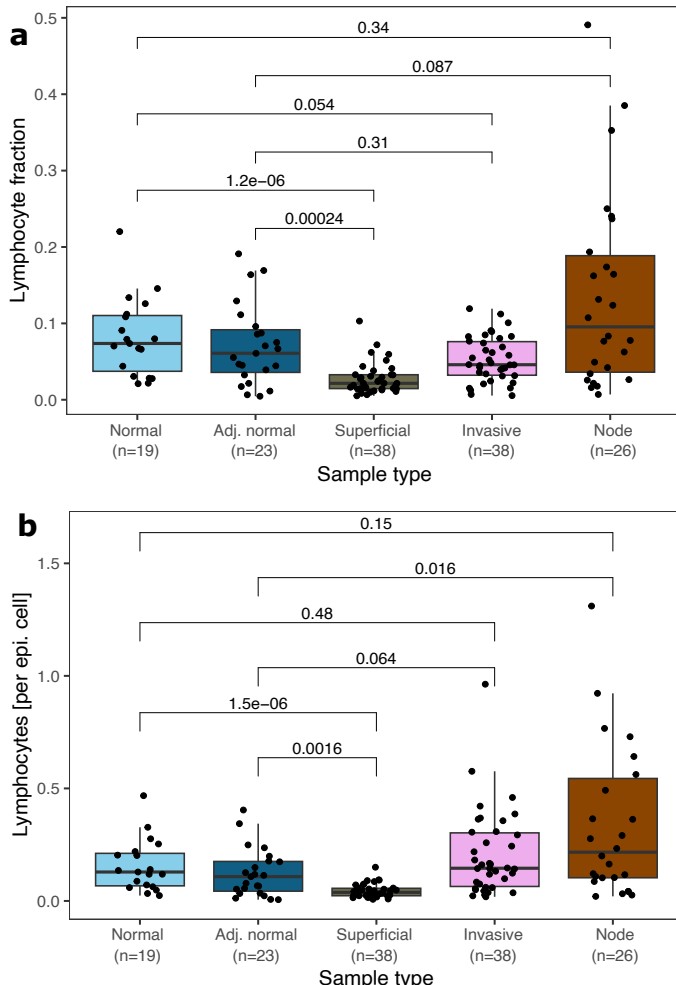

**Extended Data Fig. 5 | Lymphocyte infiltration quantified from H&E images. (a, b)** Fraction of lymphocytes (**a**) and number of lymphocytes normalised by epithelial cell count (**b**) in normal mucosa and tumour-containing regions. Lymphocyte counts were derived from H&E stained images using a deep learning-based cell classifier.

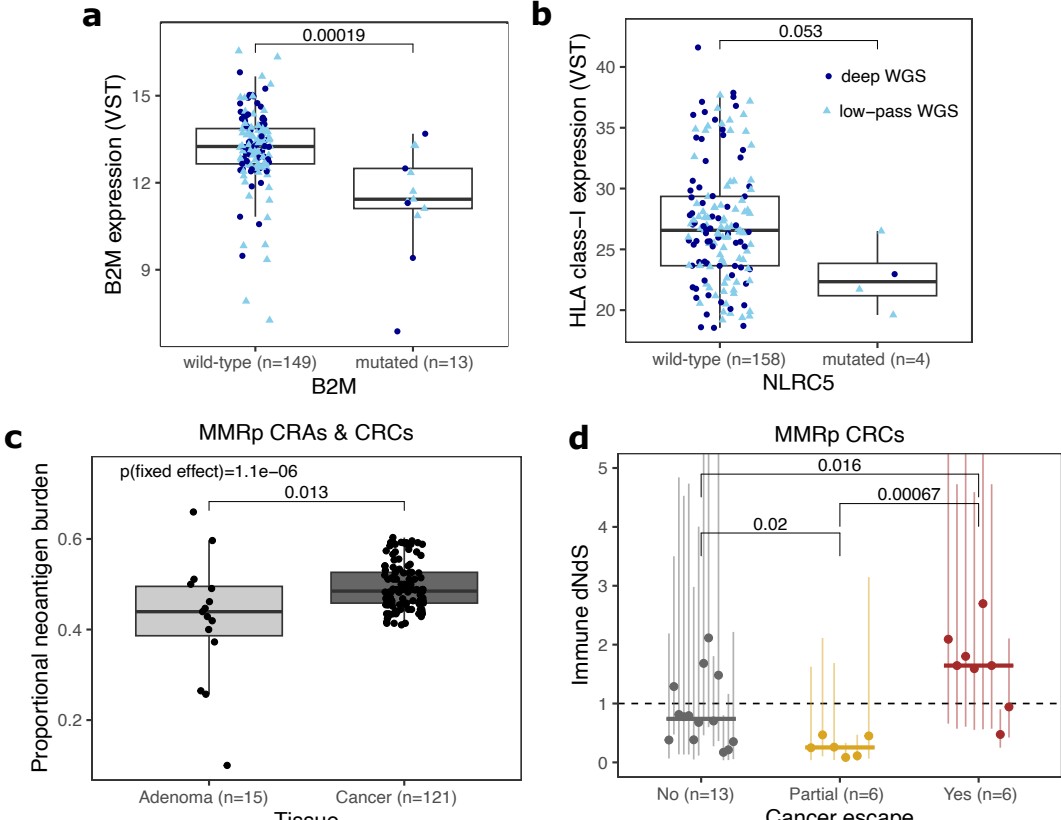

**Extended Data Fig. 6 | Relationship between immune escape status and immune measures.** (**a**) Expression of B2M in samples with and without a mutation in the gene. (**b**) Total combined expression of HLA class-I genes (HLA-A, -B and -C) compared in samples with and without a mutation in NLRC5. (**c**) Proportional neoantigen burden of MMRp-only colorectal adenomas (CRAs) and cancers (CRCs). Each dot represents a biopsy, p-value of the mixed-effects model incorporating Patient as a random effect is indicated on the top of panel. (**d**) Immune-dNdS values in MMRp FF-WGS cancers. Dots depict immune-dNdS values per biopsy, with error bars showing corresponding 95% confidence intervals.

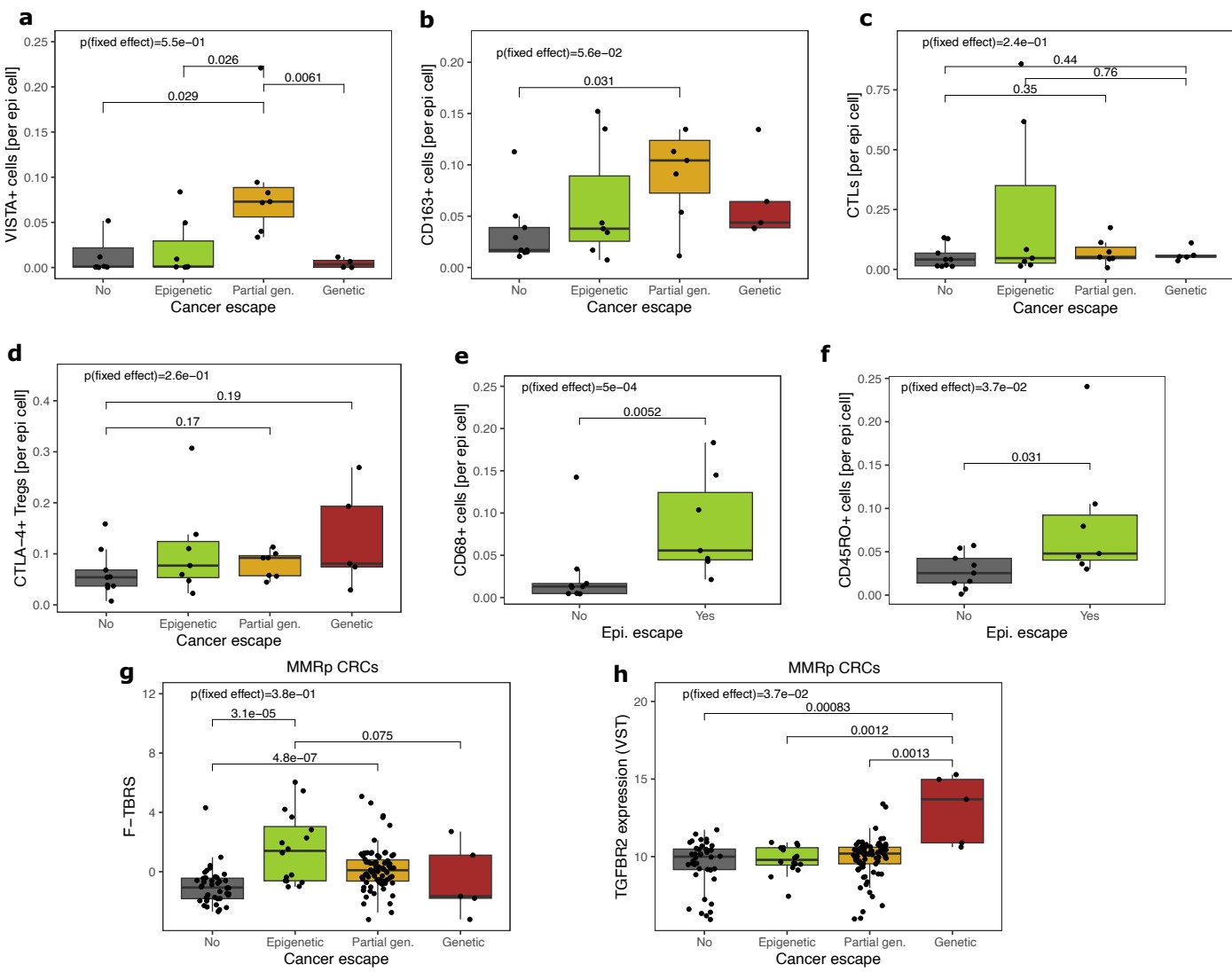

**Extended Data Fig. 7 | Relationship between immune escape status and immune infiltrates and signatures.** (a–d) Number of VISTA+ cells (a), CD163+ cells (M2 macrophages) (b), CTLs (c) and CTLA-4+ Tregs (d) per epithelial cell, shown based on cancer immune escape status. A single observation in (a) at No escape, VISTA + = 2.85 has been omitted for clarity. (e, f) Number of CD68+ cells (e) and CD45RO+ cells (f) per epithelial cell, compared between non-escaped and only epigenetically escaped cancers. Number of biopsies in a-f in each category are n(No escape)=9, n(Epigenetic escape)=7, n(Partial gen.)=7, n(Yes)=5. (g, h) Fibroblast TGFbeta signature (F-TBRS, g) and TGFBR2 (h) expression across cancers of different escape status, quantified from matched RNAseq of FF-WGS samples. Number of biopsies in g-h in each category are n(No)=40, n(Epigenetic escape)=16, n(Partial gen.)=78, n(Yes)=5. Each dot on the panels represent a biopsy and p-value of the mixed-effects model incorporating Patient as a random effect is indicated on the top of each panel.

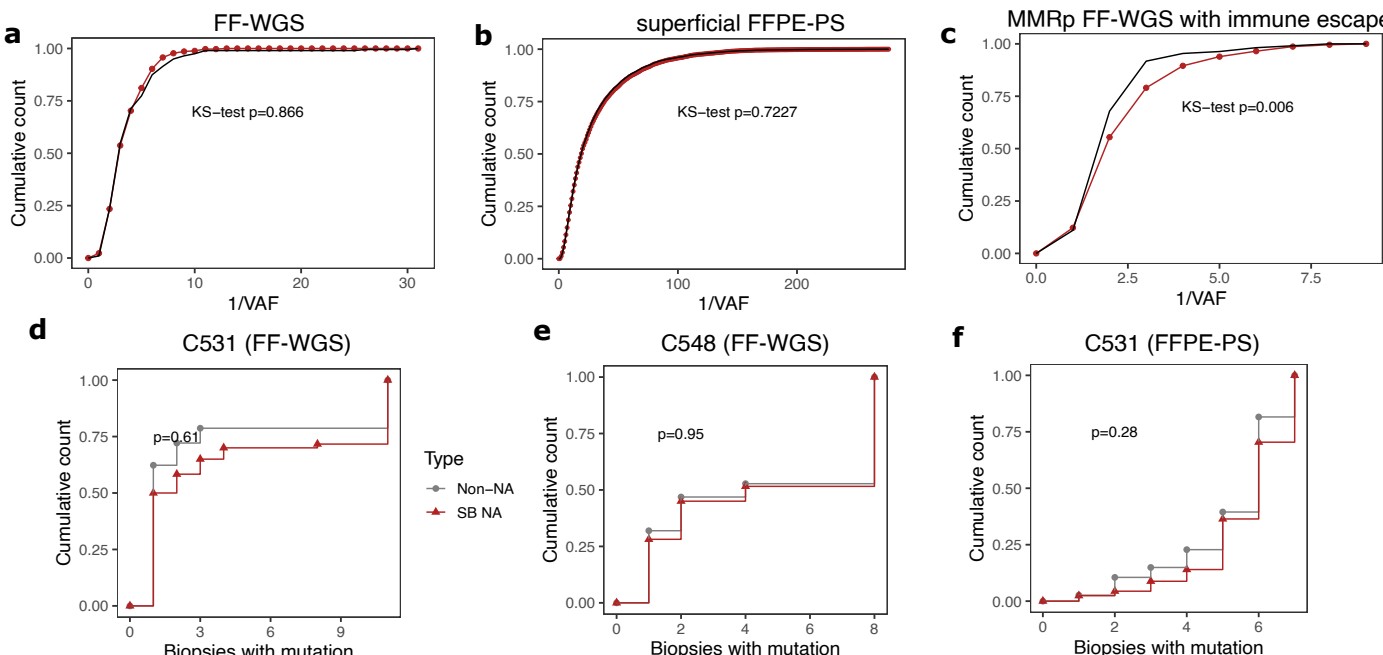

**Extended Data Fig. 8 | Frequency distribution of neoantigens.**
(**a**–**c**) Cumulative number of non-antigenic (grey) and neoantigen (red) mutations shown against the inverse of the variant allele frequency. All mutations were pulled together from fresh frozen whole genome sequencing (FF-WGS) non-escaped cancers (**a**), formalin-fixed paraffin-embedded panel sequencing (FFPE-PS) superficial tumour samples (**b**), or FF-WGS escaped MMRp samples (**c**). (**d**–**f**) The cumulative distribution of the number of mutations shared by a given number of biopsies, shown for non-neoantigen (in grey) and neoantigen (in red) mutations in the indicated cancers. Two-sided Kolmogorov-Smirnov test results are shown above the graphs.

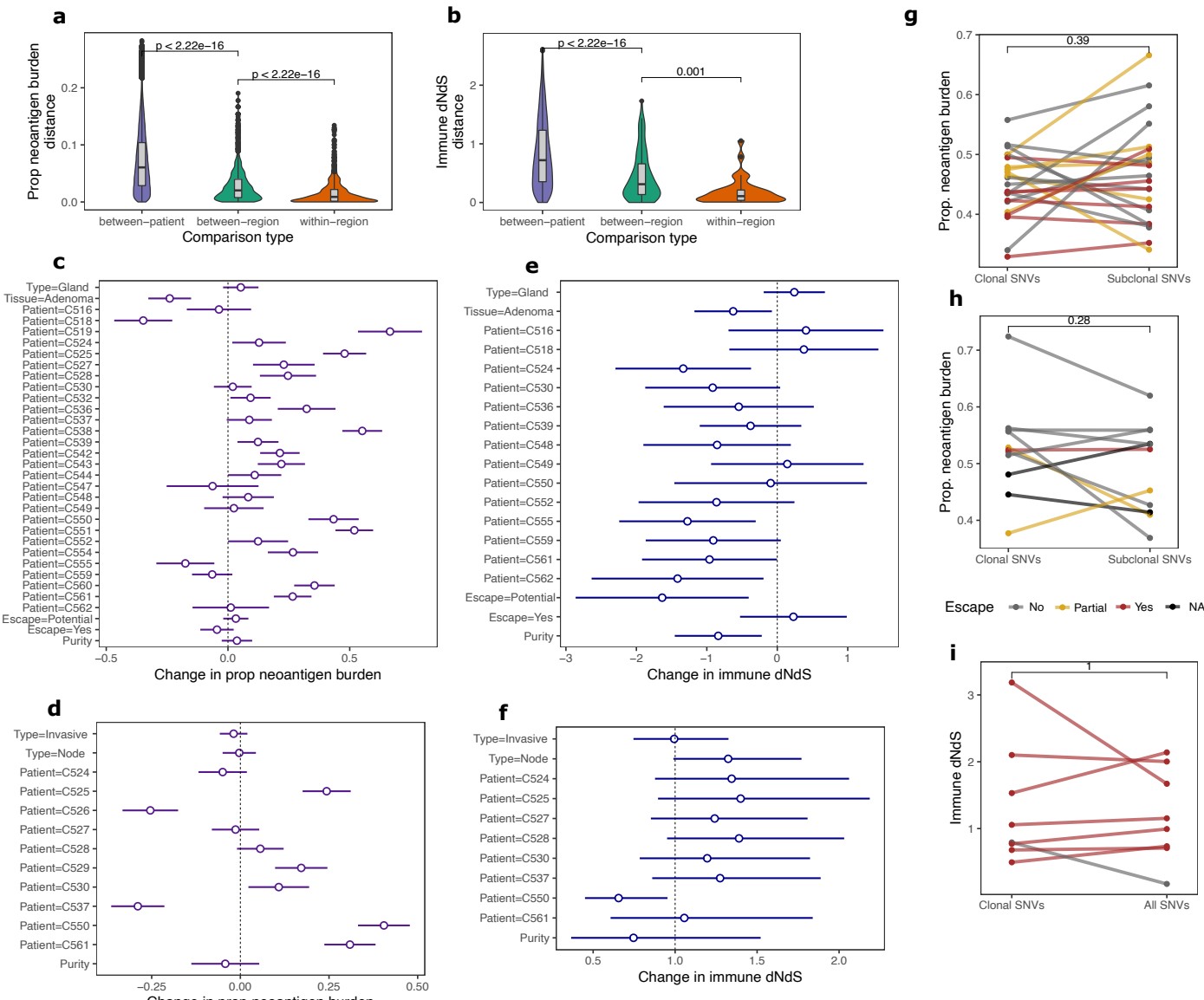

**Extended Data Fig. 9 | Intra-tumour heterogeneity of neoantigen landscape and immuno-editing in CRCs. (a, b)** Distribution of pairwise differences in proportional neoantigen burden (**a**) and immune dNdS (**b**) between fresh frozen whole genome sequencing (FF-WGS) biopsies from the same tumour region (orange), from different tumour regions of the same tumour (green) and from different cancers (purple). In total n = 470 samples were compared in (**a**) and n = 121 in (**b**). (**c–f**) Forest plots depicting the output of a multivariable regression (Methods) investigating the association of proportional neoantigen burden (**c, d**) and immune-dNdS (**e, f**) with other sample characteristics, in FF-WGS

(**c, e**, n = 495) and formalin-fixed paraffin-embedded panel sequencing (FFPE-PS) (**d, f**, n = 92) samples. Circles denote the estimated coefficients, with whiskers showing 95% confidence intervals. (**g, h**) Comparison of clonal and subclonal neoantigen burden in n = 24 FF-WGS cancers (**g**) and in n = 9 FFPE-PS cancers (**h**). Each line indicates paired values derived from the same cancer. (**i**) Comparison of immune-dNdS computed from clonal and all mutations in n = 8 FF-WGS cancers with sufficient coverage/burden to compute immune-dNdS. Each line indicates paired values derived from the same cancer.

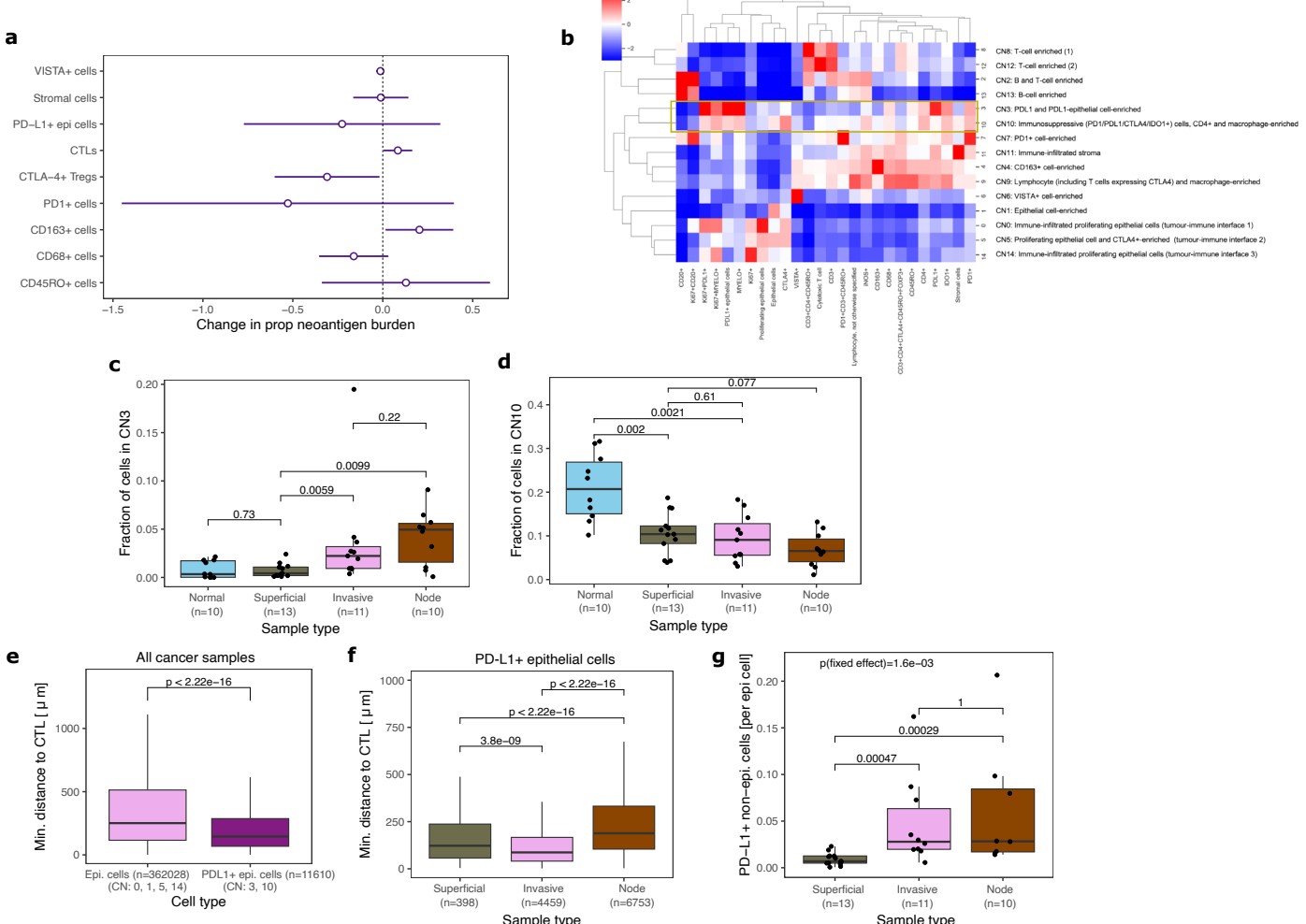

**Extended Data Fig. 10 | TME composition, PD-L1+ cellular neighbourhoods and cell-cell interactions. (a)** Forest plot depicting the output of a multivariable regression (Methods) investigating the association of proportional neoantigen burden with sample type (not plotted), patient (not plotted) and immune infiltrates in n = 28 samples. Circles denote the estimated coefficients, with whiskers showing corresponding 95% confidence intervals. **(b)** Heatmap depicting the derived cellular neighbourhoods (CN) and the relative abundance of cell types in each neighbourhood. Rows and columns are ordered according to hierarchical clustering. CN3 and CN10 are highlighted by the yellow frame.

**(c, d)** Fraction of cells in each sample type that belong to cellular neighbourhoods 3 **(b)** and 10 **(c)**. **(e)** Distance of epithelial cells to the closest cytotoxic T-cell, for non-PDL1-expressing epithelial cells in non-PD-L1+ CNs and for PD-L1+ epithelial cells in PD-L1-associated CNs, in all cancer samples. **(f)** Distance of PD-L1+ epithelial cells (from CN3 and 10) to the closest cytotoxic T-cell, shown separately for tumour sample types. **(g)** The number of PD-L1+ non-epithelial cells (per all epithelial cells) in cyclic immunofluorescence (CyCIF) images across different sample types. Each dot on **(g)** represents a biopsy, and p-value of the mixed-effects model incorporating Patient as a random effect is indicated on top.

Andrea Sottoriva
Trevor Graham

# Reporting Summary

## Statistics

For all statistical analyses, confirm that the following items are present in the figure legend, table legend, main text, or Methods section.

| n/a | Confirmed | |
|---|---|---|
| ☐ | ☒ | The exact sample size ($n$) for each experimental group/condition, given as a discrete number and unit of measurement |
| ☐ | ☒ | A statement on whether measurements were taken from distinct samples or whether the same sample was measured repeatedly |
| ☐ | ☒ | The statistical test(s) used AND whether they are one- or two-sided<br>*Only common tests should be described solely by name; describe more complex techniques in the Methods section.* |
| ☐ | ☒ | A description of all covariates tested |
| ☐ | ☒ | A description of any assumptions or corrections, such as tests of normality and adjustment for multiple comparisons |
| ☐ | ☒ | A full description of the statistical parameters including central tendency (e.g. means) or other basic estimates (e.g. regression coefficient) AND variation (e.g. standard deviation) or associated estimates of uncertainty (e.g. confidence intervals) |
| ☐ | ☒ | For null hypothesis testing, the test statistic (e.g. $F$, $t$, $r$) with confidence intervals, effect sizes, degrees of freedom and $P$ value noted<br>*Give P values as exact values whenever suitable.* |
| ☒ | ☐ | For Bayesian analysis, information on the choice of priors and Markov chain Monte Carlo settings |
| ☐ | ☒ | For hierarchical and complex designs, identification of the appropriate level for tests and full reporting of outcomes |
| ☒ | ☐ | Estimates of effect sizes (e.g. Cohen's $d$, Pearson's $r$), indicating how they were calculated |

*Our web collection on statistics for biologists contains articles on many of the points above.*

## Software and code

Policy information about availability of computer code

| Data collection | No software was used for data collection. |
|---|---|
| Data analysis | The following software were used for sequencing data and image processing: NDPViewer (v2.9.29), Fgbio (v1.3.0), Picard (v2.20.3), Mutect2 (v4.1.4.1), Platypus (v0.8.1.1), polysolver (v1.0), MuTect (v1.16) within polysolver, Strelka2 (v2.9.10), sequenza (v2.1.2), LOHHLA (https://bitbucket.org/mcgranahanlab/lohhla/src/master/), SOPRANO (https://github.com/luisgls/SOPRANO), NeoPredPipe (https://github.com/MathOnco/NeoPredPipe), bam-readcount (v1.0.1), R package lme4 (v1.1.36)<br>See Methods section of the manuscript for further detail.. All data analysis script in R (v4.4.2) creating processed data and producing figures is available from https://github.com/elakatos/EPICC_immune_analysis . |

For manuscripts utilizing custom algorithms or software that are central to the research but not yet described in published literature, software must be made available to editors and reviewers. We strongly encourage code deposition in a community repository (e.g. GitHub). See the Nature Portfolio guidelines for submitting code & software for further information.

## Data

Policy information about availability of data

All manuscripts must include a data availability statement. This statement should provide the following information, where applicable:

- Accession codes, unique identifiers, or web links for publicly available datasets
- A description of any restrictions on data availability
- For clinical datasets or third party data, please ensure that the statement adheres to our policy

Processed data used in the figures and to derive summary tables are available at Mendeley: https://doi.org/10.17632/cjfmmc95dm.2. Raw sequencing reads of FFPE-PS samples are currently made available on the European Genome-Phenome Archive (please see our cover letter for details on the delay in providing accession codes). Raw sequencing data of FF-WGS samples is available at accession code EGAS00001005230, and previously generated processed data at https://data.mendeley.com/datasets/7wx3chtsxx/2.
For alignment of sequencing reads, reference genome hg38 accessible at https://ftp.1000genomes.ebi.ac.uk/vol1/ftp/technical/reference/GRCh38_reference_genome/ was used.

## Research involving human participants, their data, or biological material

Policy information about studies with human participants or human data. See also policy information about sex, gender (identity/presentation), and sexual orientation and race, ethnicity and racism.

| | |
|---|---|
| Reporting on sex and gender | Our work used re-analysis and re-sequencing of previously collected data. All investigators were blinded to clinicopathological information and all data analysis considered the dataset as a whole without accounting for patient characteristics. |
| Reporting on race, ethnicity, or other socially relevant groupings | See above. |
| Population characteristics | Our work used re-analysis and re-sequencing of previously collected data. For information on the original data, see information reported in https://www.nature.com/articles/s41586-022-05202-1 . |
| Recruitment | See above. |
| Ethics oversight | All patients gave informed consent for collection of their materials to the UCLH Cancer Biobank (Research Ethics Committee approval 15/YH/0311). All investigators were blinded to patient data related to outcome, and all clinicopathological information. |

Note that full information on the approval of the study protocol must also be provided in the manuscript.

# Field-specific reporting

Please select the one below that is the best fit for your research. If you are not sure, read the appropriate sections before making your selection.

☒ Life sciences     ☐ Behavioural & social sciences     ☐ Ecological, evolutionary & environmental sciences

For a reference copy of the document with all sections, see nature.com/documents/nr-reporting-summary-flat.pdf

# Life sciences study design

All studies must disclose on these points even when the disclosure is negative.

| | |
|---|---|
| Sample size | No sample size calculation was performed but all available data from our previous dataset was used. For FFPE sample analysis, all patients with stage III MMRp cancer with lymph node deposit were used. Post-hoc simulations to highlight shortcomings in statistical power were performed and included in the manuscript. |
| Data exclusions | RNAseq data were filtered based on paired WGS purity (>0.05) and read count (>5M reads). WGS data were filtered based on purity (>20%). Unless stated otherwise, colorectal adenomas were excluded from all RNAseq and WGS analysis. Wherever indicated in figure legend/title, MMRd carcinomas were excluded and analysis limited to MMRp CRCs. In downstream analysis of processed images, cells with contradictory positive markers were excluded. |
| Replication | No experiments, only data extraction and analysis were carried out in the study. Repeat sampling of FF-WGS/FFPE-PS tumours served as pseudo-replicates. |
| Randomization | Our study was observational, so randomisation was not relevant for the study setting. In all analysis, purity was taken into account as covariate. In multivariable regression, sample type (gland/bulk) and purity were both included as covariates. In all analysis, adenoma vs carcinoma label was either included as a covariate or adenomas filtered out prior to analysis. |
| Blinding | All investigators were blinded to all patient clinical characteristics. |

# Reporting for specific materials, systems and methods

We require information from authors about some types of materials, experimental systems and methods used in many studies. Here, indicate whether each material, system or method listed is relevant to your study. If you are not sure if a list item applies to your research, read the appropriate section before selecting a response.

## Materials & experimental systems

| n/a | Involved in the study |
|---|---|
| ☐ | ☒ Antibodies |
| ☒ | ☐ Eukaryotic cell lines |
| ☒ | ☐ Palaeontology and archaeology |
| ☒ | ☐ Animals and other organisms |
| ☒ | ☐ Clinical data |
| ☒ | ☐ Dual use research of concern |
| ☒ | ☐ Plants |

## Methods

| n/a | Involved in the study |
|---|---|
| ☒ | ☐ ChIP-seq |
| ☒ | ☐ Flow cytometry |
| ☒ | ☐ MRI-based neuroimaging |

## Antibodies

| | |
|---|---|
| Antibodies used | Ki67 (CST, #11882S), CD8 (Fisher, #53-0008-82), CD163 (Abcam, #ab218293), CD45RO (Biolegend, #304212), CD20 (Fisher, #15301990), E-cadherin (CST, #3199S), IDO1 (CST, #10312S), CD3 (Abcam, #ab208514), CD68 (CST, #79594S), CTLA4 (Abcam, #ab283489), HLA-ABC (Fisher, #15804219), FOXP3 (Fisher, #15588936), Pan CK (antibodies online, #bs-1712R-A555) , PD1 (Abcam, #ab201825), PDL1 (Abcam, #ab267563), Vista (CST, #92734S), CD57 (Miltenyi, #130-111-964), Myeloperoxidase (Abcam, #ab252131), Vimentin (Biolegend, #677807), iNOS, CD45 |
| Validation | The specificity of all primary antibodies was validated by a pathologist using FFPE sections of colorectal cancer.  Antibodies were first tested in single-plex chromogenic immunohistochemistry, before incorporating in the multiplex CyCIF panel.  At least three dilutions of each primary antibody were tested in multiplex CyCIF, and the dilution with the brightest specific signal with no non-specific signal was selected. |

## Plants

| | |
|---|---|
| Seed stocks | *Report on the source of all seed stocks or other plant material used. If applicable, state the seed stock centre and catalogue number. If plant specimens were collected from the field, describe the collection location, date and sampling procedures.* |
| Novel plant genotypes | *Describe the methods by which all novel plant genotypes were produced. This includes those generated by transgenic approaches, gene editing, chemical/radiation-based mutagenesis and hybridization. For transgenic lines, describe the transformation method, the number of independent lines analyzed and the generation upon which experiments were performed. For gene-edited lines, describe the editor used, the endogenous sequence targeted for editing, the targeting guide RNA sequence (if applicable) and how the editor was applied.* |
| Authentication | *Describe any authentication procedures for each seed stock used or novel genotype generated. Describe any experiments used to assess the effect of a mutation and, where applicable, how potential secondary effects (e.g. second site T-DNA insertions, mosaicism, off-target gene editing) were examined.* |

