## [Peer Review File · Nature Genetics]

Epigenetically-driven and early immune evasion is key in colorectal cancer evolution

Corresponding Author: Professor Trevor Graham

Version 0:

Decision Letter:

23rd Apr 2024

Dear Professor Graham,

How are you?

First, I'm so sorry for the delay in returning this decision to you. Thank you for bearing with me.

Your Article, "Epigenome and early selection determine the tumour-immune evolutionary trajectory of colorectal cancer" has now been seen by 3 referees. Please note that Reviewers #1 and #2 reviewed your work together. You will see from their comments copied below that while they find your work of considerable potential interest, they have raised quite substantial concerns that must be addressed. In light of these comments, we cannot accept the manuscript for publication, but would be very interested in considering a revised version that addresses these serious concerns.

We hope you will find the referees' comments useful as you decide how to proceed. If you wish to submit a substantially revised manuscript, please bear in mind that we will be reluctant to approach the referees again in the absence of major revisions that address all reviewer concerns.

To guide the scope of the revisions, the editors discuss the referee reports in detail within the team, including with the chief editor, with a view to identifying key priorities that should be addressed in revision and sometimes overruling referee requests that are deemed beyond the scope of the current study. We hope that you will find the prioritised set of referee points to be useful when revising your study. Please do not hesitate to get in touch if you would like to discuss these issues further.

If you choose to revise your manuscript taking into account all reviewer and editor comments, please highlight all changes in the manuscript text file. At this stage we will need you to upload a copy of the manuscript in MS Word .docx or similar editable format.

*2) If you have not done so already please begin to revise your manuscript so that it conforms to our Article format instructions, available here. Refer also to any guidelines provided in this letter.

*3) Include a revised version of any required Reporting Summary: <https://www.nature.com/documents/nr-reporting-summary.pdf>

Please be aware of our [guidelines](https://www.nature.com/nature-research/editorial-policies/image-integrity) on digital image standards.

Link Redacted

If you wish to submit a suitably revised manuscript we would hope to receive it within 6 months. If you cannot send it within this time, please let us know. We will be happy to consider your revision so long as nothing similar has been accepted for publication at Nature Genetics or published elsewhere. Should your manuscript be substantially delayed without notifying us in advance and your article is eventually published, the received date would be that of the revised, not the original, version.

Thank you for the opportunity to review your work.

Sincerely,

Safia Danovi, PhD
Senior Editor, Nature Genetics
ORCID: 0009-0007-7822-5479

Referee expertise:

Referee #1: CRC genomics

Referee #2: CRC tumour immunology, evolution

Reviewers' Comments:

Reviewer #1:

Remarks to the Author:

Summary:

In the manuscript "Epigenome and early selection determine the tumour immune evolutionary trajectory of colorectal cancer," authors integrate a previously-collected, large-scale multiomics dataset from CRC patients' tumors with newly-collected data prioritizing a view of invasive margins to study the interaction of epigenetic reprogramming and immune evasion. The major findings are (1) epigenetic changes are one route of immune evasion through APG and/or neoantigen silencing, and (2) major immune evasion events occur early (are often clonal) and later bottlenecks have lesser effects on the immunophenotype of cancers.

This paper provides a thorough analysis of the interaction between the three aspects of immune evasion (escape vs. editing vs. exclusion). It also begins to layer on the impact of epigenetic reprogramming on these, but at a somewhat superficial level: the authors find that decreased accessibility at promoters may preferentially impact clonal neoantigens, but they do not clearly tie this in with other features of immune evasion (e.g. immune editing) as they do for genetic events. It is unclear from the current results precisely how functional these accessibility changes are, since they don't seem to generally correspond with gene expression changes. Therefore, the evidence for decreased chromatin accessibility as a means of transcriptional silencing as it currently stands is weak.

Overall, the immune editing results predominate and are quite similar to what has previously been shown in lung cancer by the TRACERx group. The paper would be improved by an expanded, integrative analysis of expression, chromatin accessibility, and spatial data. Likewise, a testable hypothesis regarding the mechanism of epigenetic silencing (e.g. a transcription factor) would be valuable.

Below I detail specific suggestions to this end, as well as other points I think would be necessary for the authors to address:

Major comments and suggestions:

1. Functional impact of somatic epigenetic alterations on immune evasion: I found it concerning that the epigenetic changes (“SCAAs”) identified in this analysis did not generally seem to correspond with expression changes. The authors suggest that this could be due to plasticity in expression, though this still raises the question of whether these changes are functional. On the other hand, perhaps more distal events (which are not accounted for in the current analysis) are compensating and/or are more meaningful than promoter-proximal events. To make the point that somatic epigenetic alterations are a means of immune evasion, the authors should demonstrate that the observed chromatin accessibility changes have functional consequences on the microenvironment. For example:
 - a. Is there a transcription factor with increased binding (based on motif enrichment) near silenced genes? If so, can it be perturbed to demonstrate transcriptional changes?
 - b. Do epigenetic alterations of APGs impact the spatial distribution of cell types in tumors, as a classic (genetic) escape mechanism might?
2. Related to point (1) above, it would be helpful to understand how expression and accessibility vary globally in this dataset, so as to better understand why the authors find SCAAs do not seem to impact expression beyond speculation on plasticity, CNVs, etc. Specifically,
 - a. Can the authors please include the global correlation between promoter accessibility and expression across the genome? We would expect this to be at least somewhat high even in a scenario with plasticity, so this would serve as an important sanity check.
 - b. Can the authors please quantify the overlap in neoantigens which are immuno-edited by epigenetic silencing vs. transcriptional silencing (Figures 2B vs. 2E)? As it stands currently, these analyses are performed separately and it is unclear whether there is any concordance. I would expect many epigenetically-silenced genes are also transcriptionally silenced, but not necessarily the other way (e.g. due to post-transcriptional modification).
3. Expanding integration of spatial and omics dimensions: Beyond demonstrating the functional impact of SCAAs, deeper integration of spatial and omics data would “close the circle” on how epigenetics influence immune evasion. Some specific suggestions are:
 - a. Similar to my point 1b above, does degree of immune surveillance (as inferred from spatial data) correlate with any of the genetic/epigenetic characteristics of the tumor? This question goes beyond characterizing genetics in the invasive margin overall (Fig. 6F) but more specifically, points toward characterizing variability dependent on presence/absence of mutations.

Minor comments suggestions:

1. Teasing apart clonality vs. statistical power: several of the results in this paper (e.g. Figures 5C-D) depend on analysis of subclonal mutations specifically, and how these results differ from those performed on clonal (or all) mutations. One potential confounder is that data will be inherently sparser for sub-clonal mutations, such that the analyses have less statistical power. To address this, I would suggest checking whether results hold if you down-sample data for clonal mutations to match the frequency of subclonal mutations, as an additional robustness test.
2. I found it difficult to get a sense of the global trends in the data from Fig. 1B without having to reference the summary in the text. For instance, it isn't very easy to inspect how correlated are dN/dS, percent neoantigen burden, etc. I would suggest supplementing Fig. 1B with an alternative visualization highlighting quantitative features; for example, 2D scatter plots of different measurements (e.g. dN/dS vs. % neoantigen burden) colored by categorical information.
3. Additional support for lymphocyte spatial patterns in H&E: can the authors please repeat the analysis describing lymphocyte distribution with CyCIF instead of relying solely on cell types inferred from H&E, i.e. by grouping all lymphocytes from the CyCIF data using a pan-lymphocyte marker?
4. Typos:
Line 198: “genes” repeated twice.
Line 396: “escape on by...”

Reviewer #2:

Remarks to the Author:

Summary:

In the manuscript “Epigenome and early selection determine the tumour immune evolutionary trajectory of colorectal cancer,” authors integrate a previously-collected, large-scale multiomics dataset from CRC patients’ tumors with newly-collected data prioritizing a view of invasive margins to study the interaction of epigenetic reprogramming and immune evasion. The major findings are (1) epigenetic changes are one route of immune evasion through APG and/or neoantigen silencing, and (2) major immune evasion events occur early (are often clonal) and later bottlenecks have lesser effects on the immunophenotype of cancers.

This paper provides a thorough analysis of the interaction between the three aspects of immune evasion (escape vs. editing vs. exclusion). It also begins to layer on the impact of epigenetic reprogramming on these, but at a somewhat superficial level: the authors find that decreased accessibility at promoters may preferentially impact clonal neoantigens, but they do not

clearly tie this in with other features of immune evasion (e.g. immune editing) as they do for genetic events. It is unclear from the current results precisely how functional these accessibility changes are, since they don't seem to generally correspond with gene expression changes. Therefore, the evidence for decreased chromatin accessibility as a means of transcriptional silencing as it currently stands is weak.

Overall, the immune editing results predominate and are quite similar to what has previously been shown in lung cancer by the TRACERx group. The paper would be improved by an expanded, integrative analysis of expression, chromatin accessibility, and spatial data. Likewise, a testable hypothesis regarding the mechanism of epigenetic silencing (e.g. a transcription factor) would be valuable.

Below I detail specific suggestions to this end, as well as other points I think would be necessary for the authors to address:

Major comments and suggestions:

1. Functional impact of somatic epigenetic alterations on immune evasion: I found it concerning that the epigenetic changes ("SCAAs") identified in this analysis did not generally seem to correspond with expression changes. The authors suggest that this could be due to plasticity in expression, though this still raises the question of whether these changes are functional. On the other hand, perhaps more distal events (which are not accounted for in the current analysis) are compensating and/or are more meaningful than promoter-proximal events. To make the point that somatic epigenetic alterations are a means of immune evasion, the authors should demonstrate that the observed chromatin accessibility changes have functional consequences on the microenvironment. For example:

a. Is there a transcription factor with increased binding (based on motif enrichment) near silenced genes? If so, can it be perturbed to demonstrate transcriptional changes?
b. Do epigenetic alterations of APGs impact the spatial distribution of cell types in tumors, as a classic (genetic) escape mechanism might?

2. Related to point (1) above, it would be helpful to understand how expression and accessibility vary globally in this dataset, so as to better understand why the authors find SCAAs do not seem to impact expression beyond speculation on plasticity, CNVs, etc. Specifically,

a. Can the authors please include the global correlation between promoter accessibility and expression across the genome? We would expect this to be at least somewhat high even in a scenario with plasticity, so this would serve as an important sanity check.

b. Can the authors please quantify the overlap in neoantigens which are immuno-edited by epigenetic silencing vs. transcriptional silencing (Figures 2B vs. 2E)? As it stands currently, these analyses are performed separately and it is unclear whether there is any concordance. I would expect many epigenetically-silenced genes are also transcriptionally silenced, but not necessarily the other way (e.g. due to post-transcriptional modification).

3. Expanding integration of spatial and omics dimensions: Beyond demonstrating the functional impact of SCAAs, deeper integration of spatial and omics data would "close the circle" on how epigenetics influence immune evasion. Some specific suggestions are:

a. Similar to my point 1b above, does degree of immune surveillance (as inferred from spatial data) correlate with any of the genetic/epigenetic characteristics of the tumor? This question goes beyond characterizing genetics in the invasive margin overall (Fig. 6F) but more specifically, points toward characterizing variability dependent on presence/absence of mutations.

Minor comments suggestions:

1. Teasing apart clonality vs. statistical power: several of the results in this paper (e.g. Figures 5C-D) depend on analysis of subclonal mutations specifically, and how these results differ from those performed on clonal (or all) mutations. One potential confounder is that data will be inherently sparser for sub-clonal mutations, such that the analyses have less statistical power. To address this, I would suggest checking whether results hold if you down-sample data for clonal mutations to match the frequency of subclonal mutations, as an additional robustness test.

2. I found it difficult to get a sense of the global trends in the data from Fig. 1B without having to reference the summary in the text. For instance, it isn't very easy to inspect how correlated are dN/dS, percent neoantigen burden, etc. I would suggest supplementing Fig. 1B with an alternative visualization highlighting quantitative features; for example, 2D scatter plots of different measurements (e.g. dN/dS vs. % neoantigen burden) colored by categorical information.

3. Additional support for lymphocyte spatial patterns in H&E: can the authors please repeat the analysis describing lymphocyte distribution with CyCIF instead of relying solely on cell types inferred from H&E, i.e. by grouping all lymphocytes from the CyCIF data using a pan-lymphocyte marker?

4. Typos:

Line 198: "genes" repeated twice.

Line 396: "escape on by..."

Reviewer #3:

Remarks to the Author:

In this study, Graham and colleagues investigate how colorectal cancer (CRC) evolves to evade the immune system. The authors analyzed tumor genetic and epigenetic changes and observed early immune evasion in CRC development. Particularly, they found immune editing and alterations in chromatin accessibility affecting antigen presentation, leading to immune escape. The evidence provided suggests that this process occurs early during the adenoma stage, whereas immune editing is negligible in CRC, except at the invasion fronts where cancer cells interact with the immune system. This data supports a "Big Bang" pattern of immune evasion, where early events define the evolution of cancer-immune interactions. I found these observations novel and of broad interest to the field. However, some parts are preliminary, the sample size is small, and some conclusions must be further supported with additional analyses. I detail my criticisms below:

1) Most of the conclusions are derived from pooled analyses of dMMR and pMMR CRC samples. Due to these two tumor types' distinct biology and immune environments, analyses should be performed separately, even considering that MMRd samples are small in number. P-values should be calculated using MMR status as a covariate in pooled analyses. Authors should clarify whether key observations hold true for both dMMR and pMMR CRCs.

2) The central conclusion of the manuscript—that immune evasion follows a "Big Bang" evolutionary pattern—is based on a single analysis of a very limited number of adenomas-CRC pairs. This aspect needs strengthening:

- I am somewhat skeptical about the notion "big bang immune evasion" applied to MMRp CRCs. According to Fig 1b, 3 out of 8 adenoma samples analyzed belong to MMRd CRCs. Besides, it appears that in two patients, adenomas are MMRp, whereas the corresponding CRC is MMRd, which may represent a significant confounding effect given the small size of the cohort. I suggest removing these two samples from the analyses.

- Authors observe significantly lower proportions of neoantigen burden in adenomas compared to CRCs. They conclude that immune escape occurs during the early stages of CRC progression based solely on this observation. However, several alternative explanations must be addressed. For instance, the lower frequency of neoantigens in adenomas may reflect distinct mutational processes (dMMR vs. pMMR; mutational signatures, etc.) than immune editing. Do adenomas have a lower mutational burden than CRCs? Could this issue impose biases in the estimation of dN/dS ratios? Moreover, did the authors check whether there is an association between the total number of mutations and neoantigen burden?

- It is important that the comparison of adenoma vs. CRC is performed individually for every patient set of samples rather than in aggregate, as shown in Fig. 4e. Does the notion of early immunoeediting hold true for all paired Ad-CRC samples or only for a subset?

- The study will benefit from including additional Ad-CRC pairs of samples in the analyses.

3) Authors briefly mention that fibroblasts are abundant in a subset of CRC that escaped immune editing. This aspect needs further exploration. Authors should assess which expression profiles correlate with escape using RNAseq data. For example, the fibroblast TGF-beta signature (F-TBRS) has been associated with poor prognosis, T-cell exclusion, and lack of responses to immunotherapy in CRC and other tumor types (Mariathasan et al. Nature 2018; Tauriello et al. Nature. 2018).

4) The authors quantify the proportion of PD-L1+ tumor cells but PD-L1 is often expressed by myeloid cells rather than by tumor cells. Did they find a correlation with PD-L1+ macrophages and the different areas – invasion fronts, core, etc.

5) For all comparisons involving individual cells, it is not reported whether cells correspond to single or multiple patients or crypts. Given that the authors attributed most of the variation observed in neoantigen burden to the originating patients, it is necessary to include both patient and crypt in the tests comparing cell characteristics. Taking individual cells as independent observations is incorrect and should be assessed with the proper statistical model (e.g., Fig4e,f).

6) When grouping genes into expression categories (line 218), did the authors check for associations with gene length?

7) In Fig.5, a summary of the patient coefficients of the model should be presented to facilitate comparison with the rest of the coefficients.

Minor:

- The processing of RNASeq data lacks description, and it is unclear how the TMP measures were obtained from the referenced publication. Moreover, TMP is not a comparable measure for expression without proper normalization and should not be used as an absolute measure of expression.

- Each comparison referred to in the text or figure should explicitly mention the test in the figure legend.

- P-values must be reported whenever a test is performed. Specifically, Fisher-test p-values and the coefficients and their confidence intervals are necessary.

- In general, the paper lacks detail, and analyses are only superficially detailed.

Version 1:

Decision Letter:

30th Jan 2025

Dear Professor Graham,

Your Article, "Epigenome and early selection determine the tumour-immune evolutionary trajectory of colorectal cancer" has now been seen by 3 referees. As before, Reviewers #1 and #2 reviewed together and You will see from their comments below that while they find your work of interest, some important points are raised. We are interested in the possibility of publishing your study in Nature Genetics, but would like to consider your response to these concerns in the form of a revised manuscript before we make a final decision on publication.

We therefore invite you to revise your manuscript taking into account all reviewer and editor comments. Please highlight all changes in the manuscript text file. At this stage we will need you to upload a copy of the manuscript in MS Word .docx or similar editable format.

*2) If you have not done so already please begin to revise your manuscript so that it conforms to our Article format instructions, available

[here](http://www.nature.com/ng/authors/article_types/index.html).

*3) Include a revised version of any required Reporting Summary: <https://www.nature.com/documents/nr-reporting-summary.pdf>

Please be aware of our [guidelines](https://www.nature.com/nature-research/editorial-policies/image-integrity) on digital image standards.

EXTENDED DATA FIGURES

Link Redacted

We hope to receive your revised manuscript within four to eight weeks. If you cannot send it within this time, please let us know.

Nature Genetics is committed to improving transparency in authorship. As part of our efforts in this direction, we are now requesting that all authors identified as 'corresponding author' on published papers create and link their Open Researcher and Contributor Identifier (ORCID) with their account on the Manuscript Tracking System (MTS), prior to acceptance. ORCID helps the scientific community achieve unambiguous attribution of all scholarly contributions. You can create and link your ORCID from the home page of the MTS by clicking on 'Modify my Springer Nature account'. For more information please visit please visit www.springernature.com/orcid.

Sincerely,

Safia Danovi, PhD
Senior Editor, Nature Genetics
ORCID: 0009-0007-7822-5479

Reviewers' Comments:

Reviewer #1 (Remarks to the Author):

Thanks very much to the authors for thoroughly considering and addressing my comments on their initial manuscript. Overall, we feel that the additions address our major comments, but we raise a few points below on clarifying the specific claims (particularly on SCAAs) and minor questions:

- a. Functional impact of SCAAs: Our initial concern centered on the functionality of these events. If expression impacts are too subtle to detect by RNA-seq (which I agree is plausible), might that also mean that they are too subtle to be functional? That said, we appreciate the author's clarification in their rebuttal that these events may contribute more to plasticity in expression than an immediate expression impact that can be observed in a "snapshot" from patients. It is also very reassuring that significant microenvironmental changes appear to result from epigenetic silencing. Our general concern will thus be alleviated so long as this "plasticity" interpretation is made clear in the text.
- b. Thanks to the authors for conducting the transcription factor analyses; it is interesting to see a few potential factors which may mitigate silencing. To be more confident in nominated factors, my outstanding question is: is the binding of proposed factors (especially NFIC) specifically enriched in SCAA loci relative to the rest of the genome? I realize there are few APG-associated SCAA, but it's important to check whether NFIC binding is ubiquitous (or, relatedly, if its motif leads to many spurious calls).
- c. While we still feel that experimental validation of TF driving immune escape in a suitable invitro co-culture or in vivo system would be of importance to connect the dots and validate the proposed mechanism, if the editor feels this is not necessary for publication in Nature Genetics, I would recommend that the authors make the limitations and hypothesis-generating nature of this work very clear in the writing and discussion.
- d. In terms of novelty and impact, I am not sure that this manuscript adds much more to what we know about immune escape beyond what the field has already learned from the TRACERx studies.
- e. Minor Point: New Figure 1C: It appears new rows were added to the heatmap but the labels weren't updated, so two rows are unannotated.

Reviewer #2 (Remarks to the Author):

Thanks very much to the authors for thoroughly considering and addressing my comments on their initial manuscript. Overall, we feel that the additions address our major comments, but we raise a few points below on clarifying the specific claims (particularly on SCAAs) and minor questions:

- a. Functional impact of SCAAs: Our initial concern centered on the functionality of these events. If expression impacts are too subtle to detect by RNA-seq (which I agree is plausible), might that also mean that they are too subtle to be functional? That said, we appreciate the author's clarification in their rebuttal that these events may contribute more to plasticity in expression than an immediate expression impact that can be observed in a "snapshot" from patients. It is also very reassuring that significant microenvironmental changes appear to result from epigenetic silencing. Our general concern will thus be alleviated so long as this "plasticity" interpretation is made clear in the text.
- b. Thanks to the authors for conducting the transcription factor analyses; it is interesting to see a few potential factors which may mitigate silencing. To be more confident in nominated factors, my outstanding question is: is the binding of proposed factors (especially NFIC) specifically enriched in SCAA loci relative to the rest of the genome? I realize there are few APG-associated SCAA, but it's important to check whether NFIC binding is ubiquitous (or, relatedly, if its motif leads to many spurious calls).
- c. While we still feel that experimental validation of TF driving immune escape in a suitable invitro co-culture or in vivo system would be of importance to connect the dots and validate the proposed mechanism, if the editor feels this is not necessary for publication in Nature Genetics, I would recommend that the authors make the limitations and hypothesis-generating nature of this work very clear in the writing and discussion.
- d. In terms of novelty and impact, I am not sure that this manuscript adds much more to what we know about immune escape beyond what the field has already learned from the TRACERx studies.
- e. Minor Point: New Figure 1C: It appears new rows were added to the heatmap but the labels weren't updated, so two rows are unannotated.

Reviewer #3 (Remarks to the Author):

The authors have addressed and clarified all my criticisms. The revised version of the manuscript has been substantially improved, including new analyses that further sustain the author's conclusions. It has also been improved in terms of readability. In my opinion, this study reveals important findings for the field and deserves to be published in Nature Genetics.

Version 2:

Decision Letter:

Our ref: NG-A64689R1

12th Mar 2025

Dear Dr Graham,

Thank you for submitting your revised manuscript "Epigenome and early selection determine the tumour-immune evolutionary trajectory of colorectal cancer" (NG-A64689R1). It has now been seen by Reviewers #1,2 who reviewed together and whose comments are below. The reviewers find that the paper has improved in revision, and therefore we'll be happy in principle to publish it in Nature Genetics, pending minor revisions to satisfy our editorial and formatting guidelines.

Sincerely,

Safia Danovi, PhD
Senior Editor, Nature Genetics
ORCID: 0009-0007-7822-5479

Reviewer #1 (Remarks to the Author):

I am satisfied with the authors' revisions and clarifications in response to reviewers' comments and find this manuscript suitable for publication.

Reviewer #2 (Remarks to the Author):

Thanks once again to the authors for their attentiveness to our comments and questions. I am satisfied that the most recent rebuttal and changes to the manuscript address my major concerns.

Reviewer #1 & #2

Remarks to the Author:

Summary:

In the manuscript “Epigenome and early selection determine the tumour immune evolutionary trajectory of colorectal cancer,” authors integrate a previously-collected, large-scale multiomics dataset from CRC patients’ tumors with newly-collected data prioritizing a view of invasive margins to study the interaction of epigenetic reprogramming and immune evasion. The major findings are (1) epigenetic changes are one route of immune evasion through APG and/or neoantigen silencing, and (2) major immune evasion events occur early (are often clonal) and later bottlenecks have lesser effects on the immunophenotype of cancers.

This paper provides a thorough analysis of the interaction between the three aspects of immune evasion (escape vs. editing vs. exclusion). It also begins to layer on the impact of epigenetic reprogramming on these, but at a somewhat superficial level: the authors find that decreased accessibility at promoters may preferentially impact clonal neoantigens, but they do not clearly tie this in with other features of immune evasion (e.g. immune editing) as they do for genetic events. It is unclear from the current results precisely how functional these accessibility changes are, since they don’t seem to generally correspond with gene expression changes. Therefore, the evidence for decreased chromatin accessibility as a means of transcriptional silencing as it currently stands is weak.

Overall, the immune editing results predominate and are quite similar to what has previously been shown in lung cancer by the TRACERx group. The paper would be improved by an expanded, integrative analysis of expression, chromatin accessibility, and spatial data. Likewise, a testable hypothesis regarding the mechanism of epigenetic silencing (e.g. a transcription factor) would be valuable.

We thank the reviewers for their insightful comments, and we are pleased that they found our analysis of immune evasion to be thorough. We appreciate the feedback on the need to better integrate the chromatin accessibility data, and to create a more integrative view, we have now included further analysis, as detailed in our replies below. Furthermore, we included potential epigenetically-driven immune escape in our summary and classification of the studied cancers (Figures 1 & 4).

Below I detail specific suggestions to this end, as well as other points I think would be necessary for the authors to address:

Major comments and suggestions:

1. Functional impact of somatic epigenetic alterations on immune evasion: I found it concerning that the epigenetic changes (“SCAAs”) identified in this analysis did not generally seem to correspond with expression changes. The authors suggest that this could be due to plasticity in expression, though this still raises the question of whether these changes are functional. On the other hand, perhaps more distal events (which are not accounted for in the current analysis) are compensating and/or are more meaningful than promoter-proximal events. To make the point that somatic epigenetic alterations are a means of immune evasion, the authors should demonstrate that the

observed chromatin accessibility changes have functional consequences on the microenvironment.

Thanks for raising this. We first note that chromatin state is “permissive” for expression, but not causative for expression. Therefore, while we might expect loss of chromatin accessibility to correlate with reduced expression, gaining of accessibility means only that transcription is possible, but does not necessarily mean that transcription is initiated. For this mechanistic reason, we should not expect tight correlations between chromatin state and gene expression. We emphasise that “plasticity” is a useful way to think about the epigenome: opening up genome accessibility is likely to increase the phenotypic states that a cell can enter (i.e. increase the range of gene expression programmes that a cell can run).

We are grateful for the specific suggestions from the reviewers on this topic, which we reply in detail to below, with specific new analyses as suggested.

For example:

a. Is there a transcription factor with increased binding (based on motif enrichment) near silenced genes? If so, can it be perturbed to demonstrate transcriptional changes?

We thank the reviewers for raising this point. We have now computationally explored which transcription factors (TFs) are predicted to bind the genomic regions in antigen presenting gene (APG) promoters that showed somatic loss of chromatin accessibility (SCAAloss), finding a small number of statistically-significant associations that should prompt further work.

We examined the list of ATACseq peaks that showed statistically significant somatic loss in at least one cancer (27 peaks, representing 20 unique APGs). We then filtered this set to peaks with loss in more than one patient (recurrent SCAAlosses), leading to a total of 10 genomic regions associated with 8 APGs. We identified TFs that are predicted to bind to these regions using the list of TF-binding sites obtained from curated human TF motifs in Heide *et al.* (2022). We found 10 TFs that bound >2 of the regions, most notably NFIC, which had putative binding sites in the silenced promoter of all 8 recurrently SCAA-affected APGs. This data is now included as Figure S2a.

When we included all APG promoter regions with SCAAloss (not only recurrent ones) in a similar manner, we again found that NFIC was predicted to bind in the promoter region of the highest number of APG promoters (13/20). RUNX1 and ZNF354C were also predicted to bind 13/20 APG promoters. An additional five TFs (SNAI2 (known as Slug), SP1, MECP2, ZFY, ZNF148) were predicted to bind >10 of the APG promoters.

Figure S2a. APGs affected by recurrent (present in >1 patient) promoter SCAAl loss and transcription factors binding ATACseq peaks within their promoter regions. TFs expressed in less than 5% of tumour samples are shown greyed out.

Of these TFs, CTCFL and TBPL2 were expressed (>1 TPM) in less than 5% of all samples, while the other TFs were expressed in most RNA sequenced samples (including NFIC), confirming their relevance in CRC. Overall, these results highlight NFIC as promising target for follow-up studies to investigate in relation to antigen presentation.

Notably, NFIC has recently been indicated in PD-L1 expression and immunotherapy response in a subgroup of lung cancer patients through its interaction with the SNP locus rs822336 (Polcaro *et al.* (2024)). This study found that silencing of NFIC led to an immune escaped phenotype.

We discuss these new results in the “Epigenetic and transcriptomic regulation of antigen presenting genes” section and in the Discussion of the manuscript.

We also note that our previous manuscript by Heide *et al.* (2022) included an analysis of somatic changes in accessibility of TF-binding sites. This analysis (carried out by computing differential accessibility between normal and cancer samples and regressing this difference against transcription start site enrichment and purity) identified genome-wide rewiring of interferon signalling associated TF-binding, suggesting suppression of pro-immune signalling (Fig. 4 of Heide *et al.* (2022)). This orthogonal analysis of all genes and all TF-binding sites (irrespective of ATACseq peaks) strengthens our claim that somatic chromatin accessibility changes play a role in shaping the tumour-immune interaction.

We appreciate that the reviewers may have been asking for direct experimentation *in vitro*. We hope the reviewers will appreciate that setting up these experiments is no small undertaking that we are, unfortunately, not suitably resourced to do. Further, *in vitro* systems of course do not have an immune component, limiting the value of *in vitro* results (i.e. we expect regulation of immune genes to be neutral in culture systems). We hope that our careful analysis of patient data is sufficient to convince of a functional effect of SCAAs.

b. Do epigenetic alterations of APGs impact the spatial distribution of cell types in tumors, as a classic (genetic) escape mechanism might?

This is an interesting suggestion: we investigated and found that SCAs at APGs are indeed associated with different tumour microenvironment composition. Specifically:

We investigated this in the 8 cancers (all MMRp) with both multiplex imaging and multi-omic data. 2/8 cancers had a SCA loss in NLRC5 (C525, 3 regions imaged; C551, 4 regions imaged). Importantly, these 2 cancers had no other genetic or epigenetic APG alteration, so we created a new category of “epigenetic” escape to classify tumours.

We then had four categories of immune escape: 3/8 cancers had no detected escape (9 regions imaged in total), 2/8 partial genetic escape (7 regions), 2/8 had epigenetic escape, and 1/8 (5 regions) had clonal high-impact genetic escape. We reiterated previously observed statistically significant differences between subtypes using this new categorisation. As a general trend, we found that epigenetically escaped cancers represented a category between non-escaped and partially genetically escaped cancers.

Figure R1 (Figs 4g-h & S7a-d). Components of the TIME compared across cancer escape categories.

Specific comparison of epigenetically escaped tumour regions to those with no escape revealed that CD68+ (macrophages) and CD45RO+ cells (memory T-cells) were enriched in the epigenetically escaped regions.

Fig S7e-f. Components of the TIME compared between cancers with no immune escape and with epigenetic (but no genetic) escape.

2. Related to point (1) above, it would be helpful to understand how expression and accessibility vary globally in this dataset, so as to better understand why the authors find SCAs do not seem to impact expression beyond speculation on plasticity, CNVs, etc.

Specifically,

a. Can the authors please include the global correlation between promoter accessibility and expression across the genome? We would expect this to be at least somewhat high even in a scenario with plasticity, so this would serve as an important sanity check.

We agree that this is an important analysis and indeed we have explored this connection in two forms: the association between (i) intra-tumour expression changes and somatic changes in accessibility; and (ii) expression and promoter accessibility levels (latter quantified as counts-per-million in ATACseq).

Point (i) was explored, together with other determinants of gene expression, in our earlier manuscripts (Heide *et al.* (2022); Househam *et al.* (2022)). In Heide *et al.* (2022). We found that over 10% of recurrent SCAs in promoter regions were associated with significantly altered expression. 24/149 (16%) of accessibility gains and 17/230 (7.5%) of losses showed clear association with expression at an FDR of 0.01. Detailed examples are shown in <https://figshare.com/articles/figure/04-RNA-seq/19857274?file=35307034>.

In Househam *et al.* (2022), we explored determinants of gene expression in a more general manner, using an eQTL-based analysis. In this analysis we found that only a quarter of the genes evaluated had expression significantly correlated with somatic genetic variations, and <4% of SNVs tested had associated changes in gene expression.

In Heide *et al.* we also found that a number of gene-expression altering SNVs co-occurred with SCAs, highlighting a mechanism through which SCAs may indirectly influence expression. (Details are shown at <https://figshare.com/articles/figure/04-RNA-seq/19857274?file=35273371>)

To address point (ii), in new analysis here prompted by the reviewers' comment, we evaluated the **correlation** between accessibility (measured as counts-per-million) at ATACseq peak locations within promoter regions (<1kb to transcription start site) and median expression (after variance stabilising transformation) of genes within normal samples and each cancer. We limited this analysis to the 11,401 genes that were found to be adequately expressed and not negatively correlated with purity in Househam *et al.* (2022). We found a weak but significant correlation in normal samples (Spearman correlation $R=0.12$). The correlation was especially pronounced for genes in group 4 (previously classified as lowly expressed genes, that were enriched for signalling pathway molecules). A potential explanation is that these genes may not be required at a high expression for essential processes, and therefore less likely to be controlled by post-transcriptional regulatory mechanisms.

We also explored each gene separately by computing the correlation coefficient between accessibility of promoter regions and median expression across cancers. We want to note that this analysis has much lower power (median number of expression-accessibility pairs in each correlation test is 22), but provides a better overview of gene-by-gene behaviour, ensuring that findings are not driven by a few genes with strong signal. We found that Spearman correlation coefficient values were significantly above 0 (Fig. R2 (left); CI: 0.069-0.077), with the highest correlation reaching $R=0.8$.

This relationship disappeared when reshuffling the cancer labels (e.g. C516) on the expression values (Fig. R2 (right)), emphasising that the observed correlation values are not driven by noise.

Figure R2. Distribution of correlation coefficients between promoter chromatin accessibility and median gene expression across patients, evaluated separately for each gene. (Left) results in the original dataset; (right) coefficients after cancer labels were shuffled (negative control). P-values of two-sided, one-sample t-tests against the null hypothesis mean=0 are shown on top of both panels.

Overall, these results suggested that (somatic) chromatin accessibility changes have a functional impact on gene expression, but due to their permissive (not causative) nature, they manifest in subtle ways.

We now provide a concise overview of this correlation analysis in the section “*Epigenetic and transcriptomic regulation of antigen presenting genes*”. In addition, the script generating the above results is available in

https://github.com/elakatos/EPICC_immune_analysis/blob/main/2.fig.epigenetic_regulation.R.

b. Can the authors please quantify the overlap in neoantigens which are immuno-edited by epigenetic silencing vs. transcriptional silencing (Figures 2B vs. 2E)? As it stands currently, these analyses are performed separately and it is unclear whether there is any concordance. I would expect many epigenetically-silenced genes are also transcriptionally silenced, but not necessarily the other way (e.g. due to post-transcriptional modification).

We would like to clarify that we define transcriptional silencing (Fig. 2e) as the silencing of a **mutant allele specifically**, without silencing of the wild-type allele. To call a neoantigen transcriptionally silenced, we require at least 10 non-mutated (and 0 mutated) reads overlapping the genomic locus. Therefore what we detect is **allele-specific expression of the wild-type allele**.

We have now evaluated the number of neoantigens that were transcriptionally and/or epigenetically silenced in at least one region of a tumour. Our analysis was limited to 71 neoantigens that had both sufficient number of RNA reads to confidently call transcriptional silencing and sufficient ATACseq information for SCAALoss calls. We found no uni-directional relationship.

	Transcr. silenced	Transcr. not silenced
Epigen. silenced	33	24
Epigen. not silenced	7	7

We do not necessarily expect a relationship between the two mechanisms of silencing, as one can imagine allele-specific expression (i) being achieved by the same allele losing chromatin accessibility (unfortunately our ATACseq data resolution is not sufficient for allele-specific chromatin accessibility detection); (ii) acting independently through RNA-level modifications.

3. Expanding integration of spatial and omics dimensions: Beyond demonstrating the functional impact of SCAAs, deeper integration of spatial and omics data would “close the circle” on how epigenetics influence immune evasion. Some specific suggestions are:

a. Similar to my point 1b above, does degree of immune surveillance (as inferred from spatial data) correlate with any of the genetic/epigenetic characteristics of the tumor? This question goes beyond characterizing genetics in the invasive margin overall (Fig. 6F) but more specifically, points toward characterizing variability dependent on presence/absence of mutations.

Prompted by the reviewers’ comment, we now explored the connection between (i) tumour regions and immune infiltrates (Fig. 3); (ii) infiltrates and genetic/epigenetic immune escape (aided the reviewers’ suggestion in point 1b, Figs. 4g-h & S7); (iii) genetic/epigenetic escape and immune selection (Figs. 4f, 5a & 5c-d); and (iv) tumour regions and immune selection (Figs. 5b & 6).

To complete this picture, we now directly explored the association between immune infiltrates and selection, focusing on intra-tumour variability. We carried out a multivariable regression to investigate how infiltrating cell counts are associated with proportional neoantigen burden, harnessing our matched sequencing and CyCIF data. We included patient as a random effect into the mixed-effect model, tumour sample type as a fixed effect (as for Fig. 5b) and additional fixed effects for immune cell phenotypes explored in Figs. 4 & S7. We found that CTLA-4+ Tregs associated negatively, while CTLs and CD163+ cells (M2 macrophages) were positively associated with proportional neoantigen burden, suggesting that tumour antigenicity is interrelated with CTL/regulatory cell infiltration, but might arise in parallel, rather than as a consequence of infiltrate levels.

Fig S11a. Association between TIME components and neoantigen burden, evaluated using a multivariable mixed-effects regression model. Patient (random effect) and sample type (fixed effect) were also included in the model, but not shown here.

Minor comments suggestions:

1. Teasing apart clonality vs. statistical power: several of the results in this paper (e.g. Figures 5C-D) depend on analysis of subclonal mutations specifically, and how these results differ from those performed on clonal (or all) mutations. One potential confounder is that data will be inherently sparser for sub-clonal mutations, such that the analyses have less statistical power. To address this, I would suggest checking whether results hold if you down-sample data for clonal mutations to match the frequency of subclonal mutations, as an additional robustness test.

We thank the reviewers for this helpful suggestion; following their advice we have now examined clonal and subclonal mutation counts, and down-sampled clonal mutation counts.

In neoantigen/non-antigenic mutation and general gene expression analysis (Fig. 2d & S2b-f) the number of clonal and subclonal mutations included are of the similar magnitude, 1976 clonal SNVs, 1384 subclonal SNVs (708 clonal FSs, 363 subclonal FSs). When subsampling to mimic S2d and S2f, we found that 68% and 74% of downsampled clonal SNV/FS datasets resulted in a significant value.

For Fig. 2e, we had substantially more (459) clonal SNVs than subclonal SNVs (53), reflecting the fact that in this case we were limited in the number of mutations with appropriate coverage of the genomic locus to establish presence/absence of the mutation. This low mutation count indeed affects our ability to detect transcriptional silencing: down-sampling clonal mutations to similar counts leads to a wide-range of OR values with 98% not significantly different from 1. We have included this additional analysis when discussing Fig. 2e and changed our Discussion to exclude any underpowered claims regarding subclonal mutation silencing.

We also explored the effect of down-sampling clonal mutations on the results in Fig. 5. We randomly down-sampled the number truncal of SNVs (regardless of neoantigen status) to 25% of the original value, thus shifting the balance in the phylogenetic trees

to derive a large portion of the signal from branches. We created 50 randomly down-sampled datasets and carried out the analyses of Fig. 5a & 5c on these. We found that multivariable regression results largely aligned with what we observed, with CRAs (as compared to CRCs) still having a significantly lower proportional neoantigen burden in the majority of the down-sampled cases, while other quantities rarely showed an association.

Replicating Fig. 5c, we found that despite the increased contributions from subclonal mutations and burden, we still did not observe significant differences between subclonally immune escaped and phylogenetically related regions. However, C543 was an exception, as in this cancer the shifted balance of the phylogenetic tree indeed did highlight an increased burden of the subclonally escaped biopsies. We think it is likely that the “mutational time” of the escape event (relatively limited number of SNVs likely to accrue post event) and the spatial sampling scheme (number of samples within vs outside subclone) determine our ability to detect (weak) effects here.

Fig S10. An example of down-sampled phylogenetic tree (a), and the output of the down-sampling analyses replicating Fig. 5a&c (b-d).

2. I found it difficult to get a sense of the global trends in the data from Fig. 1B without having to reference the summary in the text. For instance, it isn't very easy to inspect how correlated are dN/dS, percent neoantigen burden, etc. I would suggest supplementing Fig. 1B with an alternative visualization highlighting quantitative features; for example, 2D scatter plots of different measurements (e.g. dN/dS vs. % neoantigen burden) colored by categorical information.

To make the comparison of different measures and categorical information easier, we now ordered cancers in Fig. 1b and 1c according to their average neoantigen burden (averaged over all cancer biopsies).

Fig 1b. Overview of FF-WGS samples, with cancers ordered by MMR status and average neoantigen burden.

In addition, we followed the reviewers' advice and visualised immune dNdS and proportional burden as scatter plots in Fig. S2. While we found a negative correlation in MMRd FF-WGS tumours, we also included the confidence intervals associated with immune dNdS measurements to highlight that this association has high uncertainty.

Fig S2c-d. Proportional burden and immune dNdS of FF-WGS (left) and FFPE-PS (right) samples.

3. Additional support for lymphocyte spatial patterns in H&E: can the authors please repeat the analysis describing lymphocyte distribution with CyCIF instead of relying solely on cell types inferred from H&E, i.e. by grouping all lymphocytes from the CyCIF data using a pan-lymphocyte marker?

We now repeated the analysis of Fig. 6a using CyCIF-derived cell phenotypes and show that the H&E and CyCIF measures are of consistent magnitude. We note that the sample size was substantially reduced as fewer ROIs were imaged through CyCIF than analysed with H&E images.

Fig R3. Infiltrating tumour ratio across different sample types as quantified from (left) CyCIF images, and (right) H&E slides with matching CyCIF data available.

To confirm that the limited number of samples is insufficient to detect an effect, we plotted ITLR results inferred from H&E, using only the slides that had a matched CyCIF ROI. We found that differences between sample types were non-significant in both imaging modalities, while CyCIF data highlighted a substantially decreased infiltration compared to normal mucosa. In addition, we note that the analysed H&E images and matched CyCIF ROIs did not fully overlap: CyCIF images covered a maximum of 5mmx5mm region centred on biopsies sampled for genomics, while H&E slides used in the classifier captured larger areas. The discrepancy was especially pronounced in the invasive margin, where the margin located in an angle often meant CyCIF images could not capture a full 500 um on both sides of the tumour-normal interface. Therefore, in this case CyCIF images represent a subsample of the full invasive margin opposed to H&E slides analysed in our original analysis.

4. Typos:

Line 198: “genes” repeated twice.

Line 396: “escape on by...”

Thank you, we have now corrected these typos.

References:

Heide, T., Househam, J., Cresswell, G.D. et al. The co-evolution of the genome and epigenome in colorectal cancer. *Nature* 611, 733–743 (2022).

Househam, J., Heide, T., Cresswell, G.D. et al. Phenotypic plasticity and genetic control in colorectal cancer evolution. *Nature* 611, 744–753 (2022).

Polcaro, G., Liguori, L., Manzo, V. et al. rs822336 binding to C/EBP β and NFIC modulates induction of PD-L1 expression and predicts anti-PD-1/PD-L1 therapy in advanced NSCLC. *Mol Cancer* 23, 63 (2024).

Reviewer #3:

Remarks to the Author:

In this study, Graham and colleagues investigate how colorectal cancer (CRC) evolves to evade the immune system. The authors analyzed tumor genetic and epigenetic changes and observed early immune evasion in CRC development. Particularly, they found immune editing and alterations in chromatin accessibility affecting antigen presentation, leading to immune escape. The evidence provided suggests that this process occurs early during the adenoma stage, whereas immune editing is negligible in CRC, except at the invasion fronts where cancer cells interact with the immune system. This data supports a "Big Bang" pattern of immune evasion, where early events define the evolution of cancer-immune interactions. I found these observations novel and of broad interest to the field. However, some parts are preliminary, the sample size is small, and some conclusions must be further supported with additional analyses. I detail my criticisms below:

We thank the reviewer for their positive and accurate summary and are glad that they found our findings novel and of interest. We appreciate their constructive critique.

1) Most of the conclusions are derived from pooled analyses of dMMR and pMMR CRC samples. Due to these two tumor types' distinct biology and immune environments, analyses should be performed separately, even considering that MMRd samples are small in number. P-values should be calculated using MMR status as a covariate in pooled analyses. Authors should clarify whether key observations hold true for both dMMR and pMMR CRCs.

We wholeheartedly agree with the reviewer that the substantial molecular differences between MMRd and MMRp cancers necessitate separate analysis. We would like to emphasise that our FFPE-PS cohort contained only MMRp cancers, therefore all microenvironment analysis (Figures 3, 4g-h & 6) was carried out on MMRp cancers only. Similarly, Fig. 4f (immune escape and proportional burden association) was limited to MMRp cancers, and patient and MMRd/p status was included as a covariate in our multivariable regression analysis of Fig. 5a-b.

In addition, we now discuss MMRp/MMRd specific results for the analyses in Fig. 2. Briefly, we found that while limited sample size (such as the number of mutations in MMRp cancers), as expected, limited our ability to derive statistically significant results in some cases, we observed similar trends in both MMRp and MMRd cancers. When examining transcriptional regulation levels, C516, a Lynch syndrome-associated MMRd case showed substantially lower levels of transcriptomic silencing than other cancers, both MMRp and MMRd, while also contributing 67% of the total mutations on which the analysis was carried out. We therefore repeated our analysis in Fig. 2e with the exclusion of this cancer, finding that our previous results held true (Fisher's exact test in all cancers without C516: $OR(\text{neoantigen \& not expressed})=2.07[0.99-4.4]$, $OR(\text{clonal neoantigen \& not expressed})=2.88[1.2-7.0]$; Fisher's exact test in MMRp cancers without C516: $OR(\text{clonal neoantigen \& not expressed})=3.7[1.1-14.3]$).

2) The central conclusion of the manuscript—that immune evasion follows a "Big Bang" evolutionary pattern—is based on a single analysis of a very limited number of adenomas-CRC pairs. This aspect needs strengthening:

We would like to clarify what we mean by "Big Bang" cancer evolution. When we first proposed this model for cancer evolution (Sottoriva *et al.* (2015)) it was to explain that

the founding cell of the extant tumour (i.e. the most recent common ancestor (MRCA) of the tumour) contained all the driver alterations necessary for cancer growth and that within-tumour evolution was not strongly shaped by (sub)clonal selection. The name “Big Bang” was chosen to make the analogy with the founding of the universe where the initial event in universe formation (the actual Big Bang!) **set the course of future universe evolution**, and the variation in the cosmic microwave background – which for cancer we made an analogy with intra-tumour heterogeneity – is a readout of the early universe evolution. Concordantly, our description of immune evasion as “Big Bang” in CRC is intended to mean that the immune evasive phenotype (at least the most critical aspects of it) have been **established by the onset of CRC expansion**, and heterogeneity in neoantigen burdens and immune microenvironments across a tumour primarily reflect this initial event in tumour formation. This conclusion was based on our observations of (i) typically truncal high-impact immune escape alterations, (ii) immune exclusion/suppression that was universal across highly distinct regions of cancers, (iii) negligible intra-tumour differences. Importantly, the “Big Bang” claim is not necessarily that immune evasion occurs exactly at the point of transformation, but only that it is acquired at, or prior to, the cancer expansion. We think our cancer data alone strongly support this.

In the manuscript we do of course report differences in immune-genomic phenotypes between adenomas and carcinomas, lending support to the notion that the immune landscape is reshaped at the adenoma-to-carcinoma transition, however the latter is not necessary for the “Big Bang” evolutionary pattern to apply (e.g. immune evasion and selection could be established before or within the adenoma stage). We note though that in our prior work examining CRCs arising from residual precursor adenomas (so called “Haggitt lesions”) we had previously suggested that immune microenvironment remodelling occurs precisely at the point of transformation (Gatenbee *et al.* (2022)).

- I am somewhat skeptical about the notion “big bang immune evasion” applied to MMRp CRCs. According to Fig 1b, 3 out of 8 adenoma samples analyzed belong to MMRd CRCs. Besides, it appears that in two patients, adenomas are MMRp, whereas the corresponding CRC is MMRd, which may represent a significant confounding effect given the small size of the cohort. I suggest removing these two samples from the analyses.

We apologise for the confusion our terminology has caused.

Following the point above, we reiterate that “Big Bang immune evasion” description refers to the conclusion that the dominant immune evasive phenotype is established at, or before, the initiation of the cancer expansion. To reach this conclusion, we are not reliant on analysis of adenomas.

We also emphasise that there is **no clonal link** between the adenomas and cancers pairs in our data. The adenomas were incidental growth found in the colectomy specimen at resection. The cancer is not the “offspring” of the adenoma. The adenomas and cancers are independent initiations of neoplastic evolution within the same colon.

Our “big bang immune evasion” hypothesis is grounded in the observation that even in MMRp CRCs we observe high-impact immune escape clonally, while subclonal

escapes (e.g. single-biopsy HLA mutation in C537) appear to have negligible effect on the neoantigen landscape.

All analysis we present are restricted to **carcinoma** samples, except for Fig. 4e (comparison of proportional burden in CRAs vs. CRCs) and 5a-b (multivariable regression on inter- and intra-patient heterogeneity in antigenicity). In Fig. 5a-b, MMRp/d status is included in the multivariable analysis in the form of the combination of the “patient” and “escape” independent variables. The revised version of Fig. 4e reports a statistical test that accounts for repeated samples from a patient (see further details in Methods and in our answer to the corresponding comment), and we now repeated this analysis restricted to only MMRp biopsies and found similar results.

Fig S5c. Proportional burden in MMRp CRAs and CRCs. The p-value of the mixed-effects model incorporating patient as a random effect (vs no fixed effect) is indicated on the top of the panel.

In general, we agree that in the two patients who had an MMRp-adenoma & MMRd-carcinoma, the differences could be driven by the biology of MMRp vs MMRd, rather than adenoma vs cancer. We now highlight this in the text (thank you to the reviewer for prompting this).

- Authors observe significantly lower proportions of neoantigen burden in adenomas compared to CRCs. They conclude that immune escape occurs during the early stages of CRC progression based solely on this observation. However, several alternative explanations must be addressed. For instance, the lower frequency of neoantigens in adenomas may reflect distinct mutational processes (dMMR vs. pMMR; mutational signatures, etc.) than immune editing. Do adenomas have a lower mutational burden than CRCs? Could this issue impose biases in the estimation of dN/dS ratios? Moreover, did the authors check whether there is an association between the total number of mutations and neoantigen burden?

We reiterate that our conclusion that immune escape occurs in early CRC progression is based on our observation that **high-impact immune escape is fully clonal or wide-spread** across multiple distinct regions of CRCs. This conclusion is not based on the comparison between adenomas and cancer. Indeed, as the reviewer alludes to, our dataset was not assembled with that comparison in mind.

We do argue that the stronger immune selection observed in adenomas may select for the immune-escaped clone that later becomes the cancer, but our conclusion is not solely based on adenoma neoantigen burdens. Adenomas and carcinomas have similar numbers of total mutations, and the total number of SNV neoantigens is

strongly associated with mutation burden (Fig. S2a-b below). However proportional neoantigen burden is broadly uncorrelated with total SNV burden, actually showing a slight decrease as total mutation burden increased.

Fig S2a-b. Total and proportional neoantigen burden plotted against total mutation burden in all MMRp (green) and MMRd (red) CRAs and CRCs.

This analysis highlighted that there are two CRA samples (from patient C530) that had substantially lower mutation burden – likely due to the low purity of these samples (0.14 and 0.17). However, the difference in neoantigen burden between CRAs and CRCs reported in Fig. 4e was observed even after the exclusion of all samples from patient C530 from the analysis:

Fig R4. Proportional burden in all/MMRp CRAs and CRCs after the exclusion of biopsies from patient C530. The *p*-value of the mixed-effects model incorporating Patient as a random effect (vs no fixed effect) is indicated on the top of the panel.

Finally, we believe it unlikely that difference in mutational processes contributes to the observed immune differences. In our previous work (Lakatos *et al.* (2020)) we showed that MMRd and MMRp cancers did not have a significantly different proportional burden, suggesting that the mutational processes distinguishing these subtypes do not have an inherently different immunogenic potential. Moreover, Cross *et al.* (2018) identified comparable signature activities between CRAs and CRCs, suggesting that differentially immunogenic mutational processes do not underlie the observed difference in neoantigen burden.

- It is important that the comparison of adenoma vs. CRC is performed individually for every patient set of samples rather than in aggregate, as shown in Fig. 4e. Does the notion of early immunoeediting hold true for all paired Ad-CRC samples or only for a

subset?

We want to clarify that our CRAs and CRCs are **not paired** in the sense that they are not part of the same neoplastic clone (i.e. the CRC did not develop from the CRA): these are instead concurrent adenomas found proximal to the cancer at the same time-point. They typically share a negligible number of mutations with the cancer from the same patient (see truncal branches in Fig. S1h,k,u,v,w,ab; these shared mutations are probably developmental in origin). We apologise for the previous lack of clarity and we now emphasise this in the main text, in section “Prevalence and consequence of genetic immune escape”.

To test association of CRA vs. CRC tissue status independent of potential patient-specific confounders (e.g. germline differences), in Figure 5a we included CRA samples with their adenomas status as an independent variable in multivariable regression. We found that this status was associated with a significantly lower proportional neoantigen burden and immune dNdS, confirming that adenomas are typically under stronger immune selection. Specifically, in 5/8 patients we found a lower neoantigen burden in adenomas, in patient C550 neoantigen burden was significantly higher in adenoma regions, and in 2/8 patients (C542, C552) no significant difference was observed.

- The study will benefit from including additional Ad-CRC pairs of samples in the analyses.

Again, we apologise for lack of clarity concerning the nature of the adenomas and carcinomas in this study. We want to emphasise that the focus of our manuscript is CRCs and how their immuno-genomic landscape is shaped by immune interactions, thus the evolutionary processes happening in early and late cancer expansion. We agree that the adenoma-to-carcinoma transition is a fascinating area of research that we have explored to some extent by our previous manuscripts (Cross *et al.* (2018), Gatenbee *et al.* (2022)). We now edited the discussion of the manuscript to emphasise this distinction.

3) Authors briefly mention that fibroblasts are abundant in a subset of CRC that escaped immune editing. This aspect needs further exploration. Authors should assess which expression profiles correlate with escape using RNAseq data. For example, the fibroblast TGF-beta signature (F-TBRS) has been associated with poor prognosis, T-cell exclusion, and lack of responses to immunotherapy in CRC and other tumor types (Mariathasan *et al.* Nature 2018; Tauriello *et al.* Nature. 2018).

We thank the reviewer for bringing our attention to TGF- β in fibroblasts: we now evaluated the pan-fibroblast signature (F-TBRS) following the same methodology used by Mariathasan *et al.* (2018). We observed significantly higher F-TBRS in partially and epigenetically escaped MMRp cancer biopsies, compared to unescaped. Further investigating the TGF- β signalling pathway, we also found increased TGFBR2 expression in cancer biopsies with genetic immune escape. This latter finding mirrors the increased stromal cell infiltration observed in CyCIF images and highlights the functional role these fibroblasts may play in shaping the cancer-immune interaction.

Fig S7g-h. Fibroblast TGF-beta signature (left) and TGFB2 expression in FF-WGS biopsies with different cancer escape status, quantified through RNAseq.

4) The authors quantify the proportion of PD-L1+ tumor cells but PD-L1 is often expressed by myeloid cells rather than by tumor cells. Did they find a correlation with PD-L1+ macrophages and the different areas – invasion fronts, core, etc.

In our original analysis, we limited our PD-L1 analysis to epithelial cells in Figs. 6b-e & S8e-f. Following the reviewer's suggestion, we now tested other (non-epithelial) cells expressing PD-L1 and found that the invasive margin and node are enriched for these compared to the superficial areas of the tumour. We now include this finding in the supplementary material.

Fig S11g. The number of PD-L1+ non-epithelial cells (normalised to epithelial cell count) across different tumour regions.

5) For all comparisons involving individual cells, it is not reported whether cells correspond to single or multiple patients or crypts. Given that the authors attributed most of the variation observed in neoantigen burden to the originating patients, it is necessary to include both patient and crypt in the tests comparing cell characteristics. Taking individual cells as independent observations is incorrect and should be assessed with the proper statistical model (e.g., Fig4e,f).

We thank the reviewer for highlighting this limitation. Each individual data-point in our figures originated from a single crypt (or mini-bulk), with indeed multiple data-points arising from a given patient. In our revised analysis we use a mixed-effects model where we incorporate Patient as a random effect alongside the tested variable

(modelled as a fixed effect), using the R package *lmer*. Details of this analysis are stated in Methods, section “Statistical analysis”.

We found a significant difference in proportional burden between CRAs and CRCs associated with the fixed effect representing tissue type. Cancer escape status, on the other hand, did not show a significant association, likely due to limitations in power. We therefore changed our discussion of this point accordingly.

In analysing immune infiltrates, we believe that the potential variability between individual regions of interest (e.g. lymph node deposit vs invasive margin samples) is sufficient to treat these samples separately even if they originate from the same patient. In these cases (e.g. Fig. 2), we now state that individual points correspond to imaged regions of interests, and further include an overall p-value on top of each panel that corresponds to the test between two mixed-effects models both considering patient as random effect and with or without a fixed effect standing for the hypothesis variable (e.g. cancer escape).

6) When grouping genes into expression categories (line 218), did the authors check for associations with gene length?

The grouping of genes (carried out in Househam *et al.* (2022)) was performed on TPM values that are inherently normalised for gene length.

7) In Fig.5, a summary of the patient coefficients of the model should be presented to facilitate comparison with the rest of the coefficients.

We have now included this information by showing the range patient coefficients cover in the revised version of Figs. 5a and 5b.

Fig 5a-b. Association between sample characteristics and neoantigen burden, evaluated using a multivariable regression model. Patient-related coefficients are shown as a range from smallest to largest.

Minor:

- The processing of RNASeq data lacks description, and it is unclear how the TPM measures were obtained from the referenced publication. Moreover, TPM is not a comparable measure for expression without proper normalization and should not be used as an absolute measure of expression.

We used data previously processed and described in Heide *et al.* (2022), where we employed the current field-standard DESeq2 pipeline for appropriate normalisation of gene-specific RNA read counts. We now include this information in the “Processing of

sequencing data” section of the Methods, together with the corresponding analysis code (https://github.com/JacobHouseham/EPICC_transcriptome) and refer the reader to the original publication for further details.

We now revised and clarify the expression measures as follows:

- counts normalised through variance stabilizing transformation (VST) are used for comparison of a single gene across samples/ sample groups
- transcript per million (TPM) values are used when comparing multiple genes, e.g. for the definition of constitutively expressed genes and gene groups 1-4. In these use-cases, VST counts are inappropriate since they normalise each gene individually, so we chose the current standard used e.g. by EMBL-EBI (<https://ebi-gene-expression-group.github.io/atlas-faqs/atlas-faqs.html#when-i-search-in-expression-atlas-what-do-the-baseline-expression-results-show>)

- Each comparison referred to in the text or figure should explicitly mention the test in the figure legend.

- P-values must be reported whenever a test is performed. Specifically, Fisher-test p-values and the coefficients and their confidence intervals are necessary.

We have now included all relevant information in either the main text describing the findings or on the corresponding figure panel (we chose to mostly report p-values in the latter way). Note that by default, comparison between groups is carried out using a two-sided paired Wilcoxon-test (which we note in Methods), and in the interest of space we only specify test details in figure legends/text if a different test was performed.

- In general, the paper lacks detail, and analyses are only superficially detailed.

We apologise that our analyses were found to be lacking in detail, and we have now ensured we report all sequencing/bioinformatic pre-processing details regarding our FFPE-PS samples in the Methods section of this manuscript. We note that our FF-WGS samples have been previously published and thus we choose to refer the reader to these publications instead of unnecessary repetition. Details of all analyses relying on these pre-processed data-sources are then detailed in Methods and can be reproduced using the code shared at

https://github.com/elakatos/EPICC_immune_analysis.

References:

- Cross, W., Kovac, M., Mustonen, V. et al. The evolutionary landscape of colorectal tumorigenesis. *Nat Ecol Evol* 2, 1661–1672 (2018).
- Gatenbee, C.D., Baker, AM., Schenck, R.O. et al. Immunosuppressive niche engineering at the onset of human colorectal cancer. *Nat Commun* 13, 1798 (2022).
- Househam, J., Heide, T., Cresswell, G.D. et al. Phenotypic plasticity and genetic control in colorectal cancer evolution. *Nature* 611, 744–753 (2022).
- Lakatos, E., Williams, M.J., Schenck, R.O. et al. Evolutionary dynamics of neoantigens in growing tumors. *Nat Genet* 52, 1057–1066 (2020).
- Mariathasan, S., Turley, S., Nickles, D. et al. TGF β attenuates tumour response to PD-L1 blockade by contributing to exclusion of T cells. *Nature* 554, 544–548 (2018).
- Sottoriva, A., Kang, H., Ma, Z., Graham, T. A. et al. A Big Bang model of human colorectal tumor growth. *Nat Genet* 47, 209-216 (2015).

Reviewers #1 & #2 (Remarks to the Author):

Thanks very much to the authors for thoroughly considering and addressing my comments on their initial manuscript. Overall, we feel that the additions address our major comments, but we raise a few points below on clarifying the specific claims (particularly on SCAAs) and minor questions:

Thanks again to the reviewers for their careful consideration of our study, and we're glad that they feel their major comments have all been satisfactorily addressed. We respond to the remaining comments below.

a. Functional impact of SCAAs: Our initial concern centered on the functionality of these events. If expression impacts are too subtle to detect by RNA-seq (which I agree is plausible), might that also mean that they are too subtle to be functional? That said, we appreciate the author's clarification in their rebuttal that these events may contribute more to plasticity in expression than an immediate expression impact that can be observed in a "snapshot" from patients. It is also very reassuring that significant microenvironmental changes appear to result from epigenetic silencing. Our general concern will thus be alleviated so long as this "plasticity" interpretation is made clear in the text.

We appreciate that it is important to expand on this point: essentially that SCAAs are only permissive for expression. In the revised version of the manuscript, we now clarify this interpretation of the data (page 9, text in red).

b. Thanks to the authors for conducting the transcription factor analyses; it is interesting to see a few potential factors which may mitigate silencing. To be more confident in nominated factors, my outstanding question is: is the binding of proposed factors (especially NFIC) specifically enriched in SCAA loci relative to the rest of the genome? I realize there are few APG-associated SCAA, but it's important to check whether NFIC binding is ubiquitous (or, relatedly, if its motif leads to many spurious calls).

Thanks for raising this important point of verification. To address it, we have now performed a genome-wide binding site analysis for all transcription factors, rather than previously where we had only considered SCAA-loss affected APG promoter regions. We find that while TFs associated with APGs are predicted to be fairly promiscuously binding across the genome, they nevertheless do show enrichment at APG loci, including those APG loci affected by SCAAs. This is true for NFIC in particular, providing support that this is not a spurious finding.

TFs predicted to recurrently bind APG SCAAl loss peaks (Extended Data Fig. 3a) were amongst TFs with highest number of predicted binding sites across the genome, but several TFs with similarly promiscuous binding motifs had no predicted binding in APG-SCAAl loss sites or covered only a single APG (Fig. R1). In particular, transcription factors FOXL1 and ZNF32 stood out as these have high number of binding sites both genome-wide and in APG-SCAAl loss promoters. However, these sites are both concentrated on ERAP2: ZNF32 has 20 binding sites in ERAP2 but no other APG, FOXL1 has 22 in ERAP2 and 2 in NLRC5. In comparison, NFIC covers all APGs uniformly – making it likely to be a better candidate if one wanted to disrupt immune escape on multiple fronts (Table S1).

Figure R1. Total number of binding sites for all transcription factors and sites overlapping recurrently SCAALoss APG promoters. TFs identified as binding recurrently SCAALoss-affected APG promoters (Extended Data Fig. 3a) are highlighted in green. In addition, FOXL1 and ZNF32 are highlighted in grey.

Specifically: looking at all binding sites across our TFs of interest versus other TFs, we found that the proportion of APG SCAALoss sites was higher in our TF subset (one-sided Fisher-test $p=0.042$), though proportions were small in both TF sets due to the low number of loci that are both in APG promoters and recurrently affected by SCAALoss.

	APG SCAALoss bind. sites	Other bind. sites
APG-highlighted TFs	63	9231962
Other TFs	220	41845359

Since <1% of TF binding sites were located in promoter regions (of any gene) affected by SCAALoss, we also evaluated the total number of these sites within each TF set compared to APG-SCAALoss ones. This analysis reiterated that although our selected TFs do cover a large number of sites, they overall have a significantly higher proportion of these overlapping with APG promoters (one-sided Fisher-test $p=0.022$) than the rest of the TFs.

	APG SCAALoss bind. sites	Other prom. SCAALoss bind. sites
APG-highlighted TFs	63	63014
Other TFs	220	298076

In summary, we found that while these identified TFs are not strongly specific to APG SCAALoss sites, their binding is enriched in these affected promoter regions. We now highlight this feature in the article (text in red on pages 8 and 25 and Table S1) as it affects how one would design experiments/treatments targeting these TFs depending on the desired specificity.

c. While we still feel that experimental validation of TF driving immune escape in a suitable invitro co-culture or in vivo system would be of importance to connect the dots and validate the proposed mechanism, if the editor feels this is not necessary for publication in Nature Genetics, I would recommend that the authors make the

limitations and hypothesis-generating nature of this work very clear in the writing and discussion.

We now expanded the Discussion (text in red) to highlight the hypothesis generating nature of this work, and the need for future functional validation.

d. In terms of novelty and impact, I am not sure that this manuscript adds much more to what we know about immune escape beyond what the field has already learned from the TRACERx studies.

We do agree that the TRACERx studies and others in other cancer types have been seminal in demonstrating immune escape in cancer, including through DNA methylation epigenetic silencing. However, we think what our study adds is exploration (i) of chromatin-remodelling as a mechanism of epigenetic immune escape and (ii) in colorectal cancer, providing an unprecedented view of the interaction between the epigenome and the immune microenvironment in this common malignancy. We now mention this in the revised version of the manuscript (page 4, text in red).

In addition, our data offers an increased resolution compared to the seminal TRACERx studies. First, our single-gland samples enabled exploration on units of cancer that are quasi-identical genetically, ensuring that heterogeneity within a single sample is not a confounder in our analysis. Second, the high number of samples per patient, organised within physically close groups allowed us to pinpoint immune evolution (e.g. escape mutations) beyond the level of clonal/subclonal classification. While we found that subclonal evolution was negligible, we would not have been able to show this without the resolution of our dataset.

e. Minor Point: New Figure 1C: It appears new rows were added to the heatmap but the labels weren't updated, so two rows are unannotated.

Thank you for pointing out this mistake. We have now added the missing labels, "CD68+" and "Stromal cells".

Reviewer #3 (Remarks to the Author):

The authors have addressed and clarified all my criticisms. The revised version of the manuscript has been substantially improved, including new analyses that further sustain the author's conclusions. It has also been improved in terms of readability. In my opinion, this study reveals important findings for the field and deserves to be published in Nature Genetics.

We thank the reviewer again for their constructive and positive assessment of our work, and we are grateful for their support.